# Federated Distillation for Whole Slide Image via Gaussian-Mixture Feature Alignment and Curriculum Integration

Luru Jing [1]   Cong Cong [2]   Yanyuan Chen [3]   Yongzhi Cao [1]

## Abstract

Federated learning (FL) offers a promising framework for collaborative digital pathology by enabling model training across institutions. However, real-world deployments face heterogeneity arising from diverse multiple instance learning (MIL) architectures and heterogeneous feature extractors across institutions. We propose FedHD, a novel FL framework that performs local Gaussian-mixture feature alignment tailored for WSI analysis. Instead of exchanging model parameters, each client independently distills semantically rich synthetic feature representations aligned with the distribution of real WSIs. To preserve diagnostic diversity, FedHD adopts a one-to-one distillation strategy, generating a synthetic counterpart for each real slide to avoid overcompression. During federation, a curriculum-based integration strategy progressively incorporates cross-site synthetic features into local training once performance plateaus. Furthermore, an optional interpretation module reconstructs pseudo-patches from synthetic embeddings, enhancing transparency. FedHD is architecture-agnostic, privacy-preserving, and supports personalized yet collaborative training across diverse institutions. Experiments on TCGA-IDH, CAMELYON16, and CAMELYON17 show that FedHD consistently outperforms state-of-the-art federated and distillation baselines.

[1]School of Computer Science, Peking University, Beijing, China [2]Center for Health Informatics, Australian Institute of Health Innovation, Macquarie University, Sydney, NSW 2113, Australia [3]School of Data Science, University of Virginia, Charlottesville, VA, USA. Correspondence to: Yongzhi Cao <caoyz@pku.edu.cn>.

*Proceedings of the $43^{rd}$ International Conference on Machine Learning*, Seoul, South Korea. PMLR 306, 2026. Copyright 2026 by the author(s).

## 1. Introduction

Histopathology image analysis using whole slide images (WSIs) plays a central role in cancer diagnosis (Wang et al., 2024; Chauhan et al., 2025; Tang et al., 2025). Multiple Instance Learning (MIL) frameworks (Lu et al., 2021; Shao et al., 2021) have greatly advanced WSI classification. However, training robust MIL models requires large and diverse datasets, which are rarely available at a single institution. Privacy regulations further restrict data sharing, making cross-institutional collaboration essential yet challenging (Dayan et al., 2021).

Federated learning (FL) provides a natural paradigm for collaborative model training without direct data exchange (Wen et al., 2023; Guan et al., 2024; Cong et al., 2026). In real-world medical settings, institutions often differ in computational resources and modeling preferences, leading to the adoption of heterogeneous feature extractors as well as diverse MIL architectures. Such heterogeneity results in incompatible model parameter spaces across clients, posing fundamental challenges to conventional FL frameworks.

To mitigate model compatibility issues, recent federated dataset distillation (FedDD) methods (Jin et al., 2025; Jia et al., 2024; Zhao & Bilen, 2023) have been proposed, enabling clients to share synthetic datasets instead of model parameters. While this paradigm facilitates collaboration across heterogeneous models, our preliminary results show that existing FedDD approaches often yield low-informative synthetic data, leading to notable performance degradation in local MIL models. We identify two key reasons for this limitation. First, the mean-matching based distribution alignment used in prior works (Zhou et al., 2021; Zhao & Bilen, 2023; Tian et al., 2023) is ill-suited for WSIs, whose patch features follow multi-component distributions (Song et al., 2024), causing severe oversimplification of morphological complexity. Second, most methods pursue maximum compression, representing thousands of real patches with only a few synthetic samples. This strategy becomes counterproductive for WSI datasets, which are typically small in size and exhibit high intra-class diversity at the slide level. Over-compression discards fine-grained diagnostic cues, resulting in synthetic data with limited representational richness.

Accordingly, we propose FedHD, a novel dataset distillation framework tailored for federated WSI learning. At the local level, FedHD introduces an enhanced distribution alignment strategy that models the patch feature space as a mixture of Gaussians and aligns both the mean and covariance of each component between real and synthetic data, effectively capturing intra-slide heterogeneity. Moreover, to preserve representational richness, FedHD performs one-to-one feature distillation, generating a synthetic counterpart for each real slide rather than compressing multiple slides into a few examples. Furthermore, unlike pixel-level distillation, FedHD operates directly on patch embeddings, which naturally align with the MIL pipeline, improving training efficiency while maintaining semantic fidelity. To enhance interpretability of these synthetic embeddings, an on-demand interpretation module further reconstructs synthetic features into realistic pseudo-patches.

During federation, each client shares its distilled slides with the server, which aggregates and redistributes them to all participating institutions. To further bridge the gap between real and synthetic data, we adopt a curriculum learning strategy that gradually incorporates synthetic data into local training. Specifically, each client first optimizes its MIL model using real data to ensure stable convergence, then synthetic data from other clients are progressively introduced as auxiliary supervision once local performance saturates. This curriculum design allows the model to learn complementary knowledge distilled by other institutions in a controlled manner, mitigating domain shift and enhancing generalization. Taken together, FedHD establishes a new direction for feature-level dataset distillation in federated pathology. It is architecture-agnostic, and privacy-preserving, enabling personalized yet collaborative model development across heterogeneous institutions. Extensive experiments on TCGA-IDH, CAMELYON16, and CAMELYON17 demonstrate that FedHD consistently outperforms state-of-the-art federated and distillation baselines in both performance and scalability.

In summary, our contributions include: 1) a novel Gaussian-mixture feature distillation framework tailored for federated WSI learning; 2) a one-to-one feature distillation strategy that preserves diagnostic diversity by generating a synthetic counterpart for each real slide, avoiding over-compression; and 3) a curriculum-based federation strategy that progressively incorporates cross-site synthetic data into local training, enhancing generalization in a controlled manner.

## 2. Related Work

**Federated Learning.** Federated Learning enables privacy-preserving, cross-institutional collaboration (Wen et al., 2023; Li et al., 2020). Existing approaches fall into two main categories: standard federated learning (Mohri et al., 2019; Kulkarni et al., 2020), which aims to train a single high-performance global model across all clients. For example, FedAvg (McMahan et al., 2017) averages client model weights after local training; recent methods such as FedMut (Hu et al., 2024) leverage mutual knowledge distillation to improve robustness against heterogeneity, while FedImpro (Tang et al., 2024) enhances generalization by sharing similar features between clients. In contrast, personalized FL (Zhang et al., 2025; Sabah et al., 2025) seeks to tailor models to individual client data distributions. For instance, FedD3 (Song et al., 2023) learns personalized representations via disentangled dual decoders, separating global and local semantics to enable personalized prediction; FedDGM (Jia et al., 2024) leverages diffusion models and generative latents to produce client-specific synthetic features, improving performance under extreme data heterogeneity.

**Federated Learning for WSI.** FedHisto (Lu et al., 2022) integrated federated learning with weakly supervised MIL, leveraging attention mechanisms to enhance slide-level classification performance. HistoFS (Raswa et al., 2025) advanced federated MIL by introducing pseudo-bag style augmentation and RoI-aware alignment. However, these approaches typically assume homogeneous computational resources and uniform MIL architectures across clients, limiting their applicability in real-world multi-institutional settings. While recent work such as FedWSIDD (Jin et al., 2025) introduces dataset distillation for collaborative WSI analysis, it relies on distribution matching to synthesize representative slides. This simple matching results in coarse-grained approximations of feature distributions, failing to preserve the intra-slide heterogeneity.

**Dataset Distillation.** Dataset distillation (DD) is a technique for efficient knowledge condensation (Wang et al., 2025; Wei et al., 2024). Mainstream approaches can be broadly categorized into meta-model matching, gradient matching, distribution matching, and trajectory matching (Xue et al., 2025). Initial studies focused primarily on natural image datasets such as MNIST (Sucholutsky & Schonlau, 2021; Loo et al., 2023) and CIFAR-10 (Li et al., 2024; Zhang et al., 2023). In recent years, the scope of DD has gradually expanded into medical imaging. For instance, a small number of synthetic chest X-rays can retain high lesion detection performance (Li et al., 2022). Similarly, a progressive trajectory matching strategy has been proposed for medical datasets (Yu et al., 2024). Despite recent progress (Cong et al., 2024), DD in WSI analysis remains limited. WSI features exhibit multi-component distributions, and simplistic alignment methods fail to capture their morphological complexity. Moreover, excessive compression in current approaches leads to loss of fine-grained diagnostic cues, reducing the representational quality of synthetic data.

**Differences from Prior Studies.** FedHD introduces a

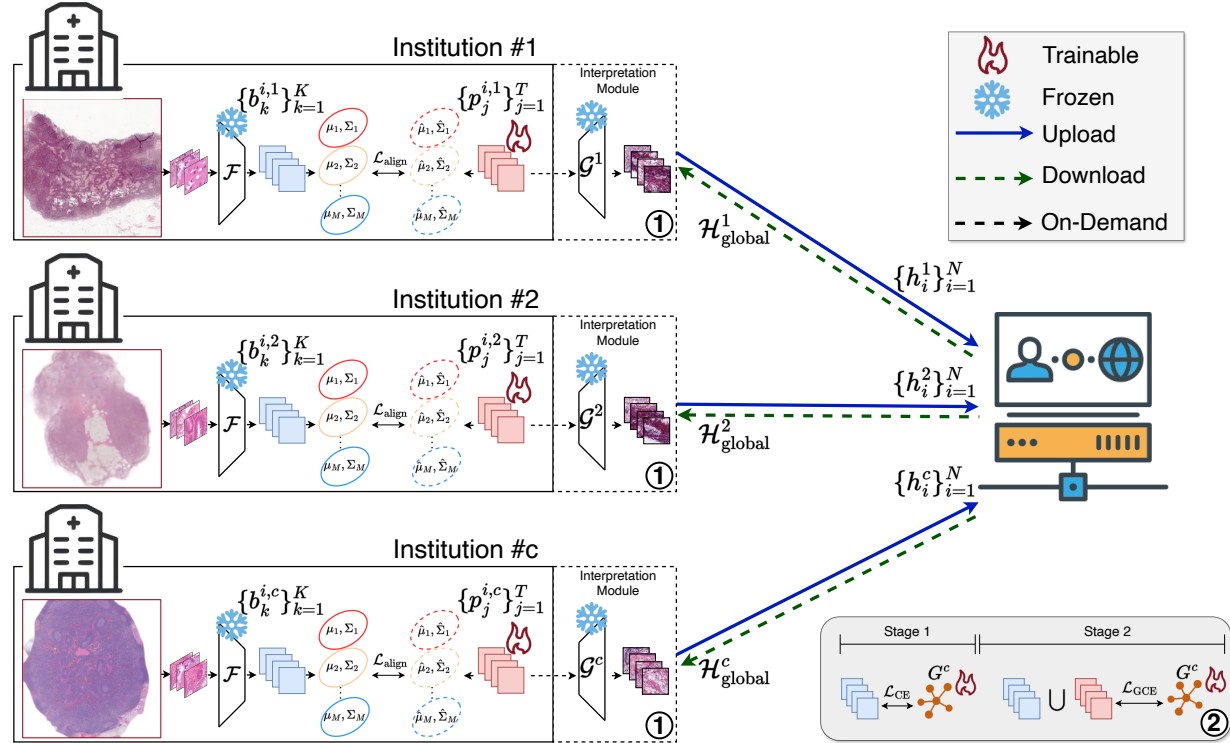

*Figure 1.* **Overview of the FedHD Framework.** ① Each institution $c$ distills its local WSIs into a set of synthetic slides ($\{h_i^c\}_{i=1}^N$) through the local Gaussian-mixture feature distillation process. $\{h_i^c\}_{i=1}^N$ are then uploaded to a central server, which aggregates them and constructs a global synthetic dataset $\mathcal{H}_{\text{global}}^{(c)}$ for each client by excluding that client's own data. ② Each institution subsequently trains its MIL model using both its local real data and the received global synthetic dataset following a curriculum learning strategy.

curriculum-based strategy that gradually incorporates synthetic data from other clients after local convergence, effectively reducing domain bias and enhancing local model performance. Unlike prior DD methods, it operates on patch embeddings rather than pixels and applies one-to-one feature distillation to preserve slide-level representational richness. An on-demand interpretation module reconstructs realistic pseudo-patches from synthetic features, enhancing interpretability. Moreover, by modeling patch features as a Gaussian mixture and aligning both mean and covariance, FedHD captures intra-slide heterogeneity more effectively than simple distribution matching.

## 3. Methodology

### 3.1. Preliminary

Suppose we have $C$ participating institutions, each maintaining its own dataset $\{D^1, D^2, \cdots, D^C\}$. Each $D^c$ contains $N$ slides represented as $\{(x_i, y_i)\}_{i=1}^N$, where $x_i$ denotes a WSI and $y_i$ is the corresponding label. Note that $N$ may vary across institutions. Each institution follows a standard WSI analysis pipeline:

1. **Patch Extraction:** Each $x_i$ is partitioned into $K$ non-

overlapping image patches $\{a_k^i\}_{k=1}^K$, where $K$ depends on the size of $x_i$.

2. **Feature Extraction:** A feature extractor $\mathcal{F}$ is applied to each instance in $\{a_k^i\}_{k=1}^K$ to generate feature embeddings $b_k^i = \mathcal{F}(a_k^i)$, where $b_k^i \in \mathbb{R}^d$.

3. **Slide-Level Aggregation:** The resulting patch-level features $\{b_k^i\}_{k=1}^K$ are aggregated using a MIL model $G$ to generate a slide-level prediction for each WSI.

Globally, in a standard FL setup, the training objective is formulated as:

$$\underset{\{G^c\}_{c=1}^C}{\arg\min} \mathcal{L} = \sum_{c=1}^C \frac{|D^c|}{|\mathcal{D}|} \mathbb{E}_{(x,y)\in D^c} \left[\mathcal{L}(G^c; x, y)\right] \quad (1)$$

where $\mathcal{D}$ denotes the union of all local datasets, and $\mathcal{L}$ is a task-specific loss function. When all institutions use the same $G^c$, the setting simplifies to homogeneous FL. However, in real-world medical environments, variations in computational infrastructure often lead to heterogeneous $G^c$. In such cases, conventional gradient or parameter sharing becomes infeasible due to architectural incompatibility, necessitating alternative approaches.

## 3.2. Overview of FedHD

FedHD addresses these limitations by shifting from parameter aggregation to feature-level dataset distillation. As shown in Fig. 1, instead of exchanging model weights, each client in FedHD generates a set of synthetic slides that capture the distribution of local WSIs without exposing raw data or model internals (Section 3.2.1). These synthetic slides are then aggregated and redistributed for collaborative training in a model-agnostic manner (Section 3.2.2).

### 3.2.1. LOCAL GAUSSIAN-MIXTURE FEATURE ALIGNMENT

To generate synthetic data without sharing raw WSIs or model parameters, FedHD performs feature-level distillation locally at each client $c$. Given a set of $N$ real slides $\{x_i^{(c)}\}_{i=1}^N$, where each slide contains $K$ patch-level features $\{b_k^{i,c}\}_{k=1}^K$, with each $b_k^{i,c} \in \mathbb{R}^d$. These features define the distribution to be distilled. To preserve intra-slide heterogeneity, we approximate the patch feature distribution of each slide using a mixture of Gaussians:

$$P_{\text{real}}^{(c,i)} \approx \sum_{m=1}^M \pi_m^{(c)} \mathcal{N}(\mu_m^{(c,i)}, \Sigma_m^{(c,i)}) \tag{2}$$

where $\mu_m^{(c,i)}$, $\Sigma_m^{(c,i)}$, and $\pi_m^{(c)}$ are the mean, covariance, and mixture weight of the $m$-th component, estimated via GMM on $\{b_k^{i,c}\}_{k=1}^K$.

For each real slide $x_i^{(c)}$, we optimize a corresponding synthetic slide $h_i^{(c)}$, containing $T$ synthetic patch embeddings ($\{p_j^{i,c}\}_{j=1}^T$), where each $p_j^{i,c} \in \mathbb{R}^d$ is initialized as learnable parameters. This forms $N$ synthetic slides per client. We model the synthetic feature distribution using:

$$P_{\text{syn}}^{(c,i)} \approx \sum_{m=1}^M \pi_m^{(c)} \mathcal{N}(\hat{\mu}_m^{(c,i)}, \hat{\Sigma}_m^{(c,i)}) \tag{3}$$

where $\hat{\mu}_m^{(c,i)}$ and $\hat{\Sigma}_m^{(c,i)}$ are computed from the synthetic features $\{p_j^{i,c}\}_{j=1}^T$, assigned to each component via the same GMM. To learn $h_i^{(c)}$, we enforce alignment between $P_{\text{real}}^{(c,i)}$ and $P_{\text{syn}}^{(c,i)}$ following the loss below:

$$\mathcal{L}_{\text{align}}^{(c,i)} = \sum_{m=1}^M \left( \|\mu_m^{(c,i)} - \hat{\mu}_m^{(c,i)}\|_2^2 + \|\Sigma_m^{(c,i)} - \hat{\Sigma}_m^{(c,i)}\|_F^2 \right) \tag{4}$$

where $\|\cdot\|_2$ denotes the Euclidean norm for vector differences, and $\|\cdot\|_F$ is the Frobenius norm used to compare covariance matrices. This one-to-one distillation strategy avoids over-compression and preserves the diagnostic diversity of each real slide. The resulting synthetic slides $\{h_i^{(c)}\}_{i=1}^N$ are retained locally for use in federation.

**On-demand Interpretation via Pseudo-Patch Reconstruction.** Although the synthetic slides $\{h_i^{(c)}\}_{i=1}^N$ preserve the diagnostic diversity of real slides, they are not inherently interpretable to human experts. To support qualitative inspection without revealing raw data, FedHD includes an optional on-demand interpretation module that reconstructs synthetic embeddings into realistic pseudo-patches.

Each client $c$ locally trains a lightweight generator $\mathcal{G}^{(c)}$ that maps a synthetic feature $p_j^{i,c}$ to a pseudo-patch $\tilde{a}_j^{i,c} = \mathcal{G}^{(c)}(p_j^{i,c})$. Here, we adopt FastGAN (Liu et al., 2021) as the backbone due to its efficiency and effectiveness in low-data regimes. The generator $\mathcal{G}^{(c)}$ and discriminator $D^{(c)}$ are trained jointly with the following adversarial loss:

$$\mathcal{L}_{\text{GAN}}^{(c)} = \mathbb{E}_{a \sim \mathcal{A}^{(c)}} \left[ \log D^{(c)}(a) \right] + \mathbb{E}_{h \sim \mathcal{P}^{(c)}} \left[ \log \left( 1 - D^{(c)}(\mathcal{G}^{(c)}(h)) \right) \right] \tag{5}$$

where $\mathcal{A}^{(c)}$ denotes the real patch distribution and $\mathcal{P}^{(c)}$ the set of synthetic features. To promote semantic fidelity, we add a patch-level reconstruction loss:

$$\mathcal{L}_{\text{rec}}^{(c)} = \mathbb{E}_{(a,h)} \left[ \|\mathcal{G}^{(c)}(h) - a\|_1 \right] \tag{6}$$

The overall interpretation objective combines both losses:

$$\mathcal{L}_{\text{interp}}^{(c)} = \mathcal{L}_{\text{GAN}}^{(c)} + \lambda_{\text{rec}} \mathcal{L}_{\text{rec}}^{(c)} \tag{7}$$

where $\lambda_{\text{rec}}$ balances realism and reconstruction accuracy.

### 3.2.2. CURRICULUM-BASED FEDERATION

After completing local feature distillation, each client $c$ transmits its synthetic slides $\{h_i^{(c)}\}_{i=1}^N$ (i.e., their distilled feature tensors) along with the corresponding slide labels to a central server. The server aggregates these slides from all clients and redistributes a global pool of synthetic slides to each participant. To ensure privacy and prevent self-training on synthetic data, each client only receives slides generated by other clients:

$$\mathcal{H}_{\text{global}}^{(c)} = \bigcup_{c' \in \{1,\ldots,C\} \setminus \{c\}} \{h_i^{c'}\}_{i=1}^N \tag{8}$$

Although the received $\mathcal{H}_{\text{global}}^{(c)}$ are aligned with corresponding real patch distributions via $\mathcal{L}_{\text{align}}$, they may still lack sufficient discriminability. To address this, FedHD adopts a curriculum strategy: each client first trains its MIL model on real data for stable convergence, then progressively incorporates synthetic data from other clients. To further mitigate potential label noise in cross-client synthetic features, we replace standard cross-entropy with the Generalized Cross-Entropy (GCE) loss (Zhang & Sabuncu, 2018):

$$\mathcal{L}_{\text{GCE}}(p, y) = \frac{1 - p_y^q}{q}, \quad q \in (0, 1] \tag{9}$$

where $p_y$ is the predicted probability for class $y$, and $q$ controls the robustness–accuracy trade-off. When $q \to 1$, GCE reduces to MAE; when $q \to 0$, it approaches CE. The overall training objective at client $c$ is:

$$\mathcal{L}_{\text{local}}^{(c)} = \mathcal{L}_{\text{real}}^{(c)} + \mathcal{L}_{\text{GCE}}^{(c)} \cdot \mathbb{I}(t \geq t_0) \qquad (10)$$

where $t$ is the current training epoch, and $t_0$ denotes the curriculum threshold. The indicator function $\mathbb{I}(t \geq t_0)$ ensures that synthetic data is only used once the model has reached a stable baseline on real data. This design allows FedHD to incorporate $\mathcal{H}_{\text{global}}^{(c)}$ in a controlled manner, improving generalization without sacrificing model stability.

# 4. Experiments

## 4.1. Datasets

**CAMELYON16** (Bejnordi et al., 2017) This breast cancer dataset involves a binary classification task distinguishing between normal and tumor slides. It comprises 399 WSIs collected from two medical centers: RUMC (C1) and UMCU (C2). RUMC contributed 169 training slides (99 normal / 70 tumor) and 74 testing slides (50 normal / 24 tumor). UMCU provided 101 training slides (60 normal / 41 tumor) and 55 testing slides (31 normal / 24 tumor).

**CAMELYON17** (Bandi et al., 2019) This dataset focuses on classifying breast cancer metastases into four categories: negative (neg), isolated tumor cells (itc), micro-metastases (micro), and macro-metastases (macro). Each of the five centers (C1–C5) contributed 20 patients, with five slides per patient. The class distributions per center are: C1: 11 itc, 10 micro, 15 macro, and 64 neg; C2: 7 itc, 23 micro, 12 macro, and 58 neg; C3: 2 itc, 8 micro, 15 macro, and 75 neg; C4: 8 itc, 13 micro, 19 macro, and 60 neg; and C5: 8 itc, 5 micro, 26 macro, and 61 neg.

**TCGA-IDH** (Liu et al., 2020) This dataset focuses on IDH mutation status classification in gliomas and includes 1016 WSIs from eight centers. Only centers contributing over 40 slides were included. Each slide is labelled as either IDH wildtype (WT) or mutant (MU), based on immunohistochemistry and/or genetic sequencing. The breakdown by center is as follows: C1: 292 WT / 119 MU, C2: 52 WT / 24 MU, C3: 8 WT / 40 MU, C4: 4 WT / 44 MU, C5: 57 WT / 61 MU, C6: 38 WT / 130 MU, C7: 3 WT / 45 MU, and C8: 84 WT / 15 MU.

## 4.2. Implementation Details

Patches of size $256 \times 256$ were sampled at $40\times$ magnification for CAMELYON16 and CAMELYON17, and at $10\times$ for TCGA-IDH. For federated learning, we adopted a single-round communication protocol: each client performs 1000 iterations of dataset distillation locally, uploads its distilled slides once, and then performs 50 epochs of local training after receiving the global synthetic set. In this protocol, client $c$ uploads $N$ synthetic slides, each containing $T$ distilled patch embeddings in $\mathbb{R}^d$, resulting in a payload of $O(NTd)$ floating-point values per client (plus labels). Synthetic features were initialized randomly with $T = 1000$, and we set the number of Gaussian mixture components to $M = 16$, following empirical evidence from (Song et al., 2024) indicating that this value provides a good approximation of morphological feature distributions. The mixture weights were set uniformly as $\pi_m = 1/M = 1/16$. In addition, we used $q = 0.7$ for the GCE loss, and set the curriculum threshold to $t_0 = 30$. Sensitivity analyses for these hyper-parameters are provided in the Appendix. We report test accuracy and Matthews Correlation Coefficient (MCC) for each client and their weighted global averages. For CAMELYON16, as an official training and test split is provided, we evaluate model performance using repeated testing with five different random seeds to account for stochastic variation during training. For CAMELYON17 and TCGA-IDH, we conduct five-fold cross-validation within each center. To prevent data leakage, slides from the same patient are always assigned to the same fold. In addition, we ensure that all classes are represented in both the training and test splits for each fold. All results are reported as mean $\pm$ standard deviation, and training was implemented in PyTorch and conducted on a single NVIDIA A100 GPU.

## 4.3. Comparison with Prior Art

**Heterogeneous Local Model** To simulate a realistic heterogeneous federated learning scenario, we consider client heterogeneity arising from both feature extractors and MIL architectures. Specifically, we construct a feature extractor pool consisting of a ResNet50 pretrained on ImageNet and three pathology foundation models, including UNI (Chen et al., 2024), PhikonV2 (Filiot et al., 2024), and GPFM (Ma et al., 2026). In parallel, we build a MIL model pool by selecting three widely used architectures in digital pathology, namely CLAM (Lu et al., 2021), TransMIL (Shao et al., 2021), and ACMIL (Zhang et al., 2024). Each participating institution randomly selects one feature extractor and one MIL model from the corresponding pools, resulting in heterogeneous local model configurations across clients.

**Performance under Heterogeneous Local Models** We compare FedHD against state-of-the-art personalized FL methods, including FedHE (Chan & Ngai, 2021), DESA (Huang et al., 2024), FedDGM (Jia et al., 2024), HistoFS (Raswa et al., 2025), and FedWSIDD (Jin et al., 2025). As shown in Table 1, FedHD consistently achieves the best performance across all benchmarks under this heterogeneous setting. For instance, on CAMELYON16,

*Table 1.* Heterogeneous model performance comparison of FedHD against competing personalized federated learning methods. Client-specific heterogeneous configurations, including feature extractors and MIL architectures, are shown in [...]. Best results are highlighted in **bold**, and underlined values indicate statistically significant improvements of FedHD over the strongest baseline ($p < 0.05$).

| | Methods | FedHE Acc | FedHE MCC | DESA Acc | DESA MCC | FedDGM Acc | FedDGM MCC | HistoFS Acc | HistoFS MCC | FedWSIDD Acc | FedWSIDD MCC | FedHD (Ours) Acc | FedHD (Ours) MCC |
|---|---|---|---|---|---|---|---|---|---|---|---|---|---|
| CAM16 | C1 [R50+CLAM] | $72.7_{\pm1.3}$ | $49.2_{\pm3.8}$ | $77.0_{\pm0.8}$ | $46.7_{\pm3.1}$ | $77.0_{\pm1.6}$ | $53.0_{\pm2.0}$ | $82.4_{\pm1.1}$ | $58.4_{\pm2.2}$ | $83.7_{\pm0.8}$ | $63.5_{\pm1.5}$ | $\underline{\mathbf{85.1_{\pm1.0}}}$ | $\underline{\mathbf{65.1_{\pm3.0}}}$ |
| | C2 [UNI+TrMIL] | $77.7_{\pm2.0}$ | $57.4_{\pm2.2}$ | $86.2_{\pm2.1}$ | $71.9_{\pm5.3}$ | $87.8_{\pm1.2}$ | $78.0_{\pm0.6}$ | $91.3_{\pm1.0}$ | $81.6_{\pm3.6}$ | $93.2_{\pm1.7}$ | $86.8_{\pm3.1}$ | $\underline{\mathbf{95.8_{\pm0.9}}}$ | $\underline{\mathbf{91.8_{\pm1.6}}}$ |
| | Avg | $75.2_{\pm3.1}$ | $53.3_{\pm5.3}$ | $81.9_{\pm4.8}$ | $60.5_{\pm12.7}$ | $83.4_{\pm4.9}$ | $65.2_{\pm13.6}$ | $86.7_{\pm4.9}$ | $69.9_{\pm12.6}$ | $88.7_{\pm4.9}$ | $75.3_{\pm12.3}$ | $\underline{\mathbf{91.2_{\pm5.0}}}$ | $\underline{\mathbf{80.6_{\pm12.0}}}$ |
| CAM17 | C1 [UNI+CLAM] | $72.3_{\pm2.6}$ | $35.9_{\pm3.9}$ | $72.3_{\pm2.6}$ | $41.1_{\pm5.8}$ | $74.3_{\pm1.7}$ | $45.3_{\pm0.7}$ | $75.9_{\pm2.1}$ | $48.3_{\pm4.3}$ | $77.3_{\pm3.6}$ | $47.4_{\pm4.7}$ | $\underline{\mathbf{83.6_{\pm2.4}}}$ | $\underline{\mathbf{61.8_{\pm9.2}}}$ |
| | C2 [R50+TrMIL] | $66.5_{\pm8.5}$ | $37.9_{\pm6.3}$ | $65.0_{\pm6.2}$ | $35.3_{\pm3.2}$ | $65.5_{\pm7.6}$ | $34.6_{\pm4.5}$ | $67.5_{\pm8.5}$ | $35.4_{\pm5.0}$ | $66.5_{\pm7.2}$ | $32.4_{\pm6.1}$ | $\mathbf{74.0_{\pm5.5}}$ | $\mathbf{43.4_{\pm6.6}}$ |
| | C3 [R50+ACMIL] | $77.0_{\pm4.2}$ | $45.3_{\pm7.7}$ | $78.0_{\pm2.7}$ | $48.4_{\pm2.1}$ | $79.0_{\pm2.2}$ | $50.8_{\pm4.9}$ | $79.0_{\pm2.2}$ | $51.6_{\pm5.4}$ | $79.0_{\pm4.2}$ | $51.1_{\pm2.4}$ | $\mathbf{84.0_{\pm4.2}}$ | $\mathbf{60.2_{\pm5.5}}$ |
| | C4 [PhV2+TrMIL] | $73.7_{\pm2.1}$ | $45.2_{\pm8.8}$ | $78.3_{\pm3.0}$ | $55.9_{\pm6.6}$ | $79.9_{\pm2.6}$ | $59.8_{\pm4.7}$ | $82.3_{\pm1.8}$ | $62.1_{\pm4.6}$ | $80.7_{\pm1.5}$ | $63.3_{\pm4.4}$ | $\underline{\mathbf{84.5_{\pm1.2}}}$ | $\underline{\mathbf{70.2_{\pm2.0}}}$ |
| | C5 [GPFM+CLAM] | $77.6_{\pm2.0}$ | $55.5_{\pm7.4}$ | $80.0_{\pm3.2}$ | $61.0_{\pm3.8}$ | $81.6_{\pm1.8}$ | $62.2_{\pm3.8}$ | $82.4_{\pm2.0}$ | $65.6_{\pm4.0}$ | $82.4_{\pm2.3}$ | $65.9_{\pm6.3}$ | $\underline{\mathbf{87.2_{\pm1.8}}}$ | $\underline{\mathbf{75.8_{\pm2.8}}}$ |
| | Avg | $73.4_{\pm4.3}$ | $44.0_{\pm8.1}$ | $74.7_{\pm6.4}$ | $48.3_{\pm9.9}$ | $76.1_{\pm6.4}$ | $50.5_{\pm10.8}$ | $77.3_{\pm6.2}$ | $52.7_{\pm11.2}$ | $77.2_{\pm6.6}$ | $52.0_{\pm13.2}$ | $\underline{\mathbf{82.7_{\pm4.8}}}$ | $\underline{\mathbf{62.3_{\pm11.3}}}$ |
| IDH | C1 [R50+CLAM] | $76.4_{\pm0.7}$ | $41.1_{\pm1.6}$ | $79.0_{\pm0.7}$ | $50.3_{\pm1.0}$ | $79.0_{\pm0.7}$ | $50.3_{\pm1.0}$ | $80.0_{\pm0.7}$ | $51.7_{\pm1.2}$ | $79.5_{\pm0.0}$ | $50.8_{\pm0.0}$ | $\underline{\mathbf{81.3_{\pm0.8}}}$ | $\underline{\mathbf{54.2_{\pm2.0}}}$ |
| | C2 [GPFM+TrMIL] | $76.0_{\pm3.5}$ | $48.0_{\pm9.6}$ | $77.3_{\pm3.5}$ | $55.1_{\pm1.6}$ | $77.3_{\pm3.5}$ | $55.1_{\pm1.6}$ | $78.7_{\pm2.9}$ | $64.1_{\pm3.8}$ | $81.3_{\pm2.7}$ | $67.0_{\pm2.0}$ | $\underline{\mathbf{85.4_{\pm3.0}}}$ | $\underline{\mathbf{73.7_{\pm4.3}}}$ |
| | C3 [R50+TrMIL] | $80.0_{\pm5.0}$ | $20.0_{\pm27.4}$ | $82.2_{\pm6.1}$ | $50.0_{\pm0.0}$ | $82.2_{\pm6.1}$ | $50.0_{\pm0.0}$ | $80.0_{\pm5.0}$ | $50.0_{\pm0.0}$ | $82.2_{\pm6.1}$ | $53.2_{\pm7.2}$ | $\underline{\mathbf{86.7_{\pm5.0}}}$ | $\underline{\mathbf{56.4_{\pm8.5}}}$ |
| | C4 [UNI+ACMIL] | $77.9_{\pm2.3}$ | $37.3_{\pm5.3}$ | $81.0_{\pm3.2}$ | $49.0_{\pm7.6}$ | $82.1_{\pm3.0}$ | $50.2_{\pm6.3}$ | $83.1_{\pm2.4}$ | $54.8_{\pm5.1}$ | $79.0_{\pm2.3}$ | $42.9_{\pm7.8}$ | $\underline{\mathbf{83.1_{\pm2.4}}}$ | $\underline{\mathbf{53.6_{\pm8.1}}}$ |
| | C5 [PhV2+TrMIL] | $71.3_{\pm2.4}$ | $45.4_{\pm7.4}$ | $68.7_{\pm2.0}$ | $43.2_{\pm4.0}$ | $68.7_{\pm2.0}$ | $43.2_{\pm4.0}$ | $73.0_{\pm1.9}$ | $46.9_{\pm1.7}$ | $76.5_{\pm2.3}$ | $53.4_{\pm5.0}$ | $\underline{\mathbf{80.9_{\pm2.4}}}$ | $\underline{\mathbf{62.2_{\pm4.5}}}$ |
| | C6 [GPFM+CLAM] | $72.0_{\pm1.3}$ | $26.0_{\pm4.0}$ | $74.5_{\pm1.8}$ | $27.6_{\pm4.1}$ | $74.5_{\pm1.8}$ | $27.6_{\pm4.1}$ | $76.4_{\pm1.3}$ | $29.4_{\pm4.0}$ | $80.0_{\pm1.5}$ | $33.8_{\pm3.2}$ | $\underline{\mathbf{82.9_{\pm1.6}}}$ | $\underline{\mathbf{48.9_{\pm3.4}}}$ |
| | C7 [R50+TrMIL] | $77.8_{\pm0.0}$ | $40.0_{\pm22.4}$ | $77.8_{\pm0.0}$ | $50.0_{\pm0.0}$ | $77.8_{\pm0.0}$ | $50.0_{\pm0.0}$ | $84.7_{\pm5.5}$ | $53.2_{\pm7.2}$ | $80.0_{\pm5.0}$ | $40.0_{\pm22.4}$ | $\underline{\mathbf{84.7_{\pm5.5}}}$ | $\underline{\mathbf{59.7_{\pm8.0}}}$ |
| | C8 [PhV2+CLAM] | $80.0_{\pm2.4}$ | $26.0_{\pm3.5}$ | $77.9_{\pm2.3}$ | $23.9_{\pm2.7}$ | $77.9_{\pm2.3}$ | $23.9_{\pm2.7}$ | $82.1_{\pm2.7}$ | $29.1_{\pm4.4}$ | $85.3_{\pm2.3}$ | $34.5_{\pm5.3}$ | $\underline{\mathbf{88.4_{\pm2.4}}}$ | $\underline{\mathbf{46.8_{\pm5.8}}}$ |
| | Avg | $76.9_{\pm3.3}$ | $35.5_{\pm10.4}$ | $78.9_{\pm4.0}$ | $44.9_{\pm10.3}$ | $79.2_{\pm4.0}$ | $44.9_{\pm10.3}$ | $80.9_{\pm4.3}$ | $47.4_{\pm9.7}$ | $80.5_{\pm4.2}$ | $47.0_{\pm10.1}$ | $\underline{\mathbf{84.8_{\pm4.1}}}$ | $\underline{\mathbf{57.0_{\pm8.4}}}$ |

FedHD attains an average accuracy of 91.2%, outperforming the strongest competing personalized FL method (FedWSIDD) by +2.5%. Moreover, a paired $t$-test conducted between FedHD and the second-best method across five runs confirms that these improvements are statistically significant, with $p$-values $< 0.05$ on most individual centers. Moreover, FedHD also demonstrates consistent client-wise improvements. Compared to WSI-specific FL approaches such as HistoFS and FedWSIDD, FedHD delivers clear gains, indicating that sharing distilled synthetic samples with enhanced representational quality provides a more effective mechanism for cross-center knowledge transfer.

**Communication and Training Efficiency** We take the largest client in TCGA-IDH (C1, 313 training slides) as an illustrative case, image-based dataset distillation methods such as DESA and FedWSIDD transmit synthetic patch images of size $10\times100\times3\times64\times64$, corresponding to approximately 49.2 MB per communication round assuming standard 32-bit floating-point representation. In contrast, FedHD supports two configurations: without the O2O strategy, it distills 10 fixed synthetic slides per client ($10 \times 1000 \times 1024$, approximately 39 MB); with O2O enabled, one synthetic slide is generated per real training slide, increasing the payload to approximately 1.19 GB. Regarding training time, on a single NVIDIA A100 GPU, distilling

synthetic patch images for 1000 rounds typically requires 10–12 hours, whereas FedHD, which distills compact feature representations, completes the same number of rounds in approximately 1 hour.

### 4.4. Ablation Studies

We evaluate the impact of the following key modules: Feature-level Data Distillation (FDD), One-to-One Distillation Strategy (O2O), Gaussian-Mixture Feature Alignment (GMA), On-Demand Interpretation, and Curriculum-Based Federation (CBF). We compare against a baseline method (Jin et al., 2025) that performs patch-level image distillation optimized for maximum data compression using naive distribution matching. Specifically, each client generates synthetic image data of size $10\times100\times3\times64\times64$, corresponding to 100 synthetic samples per slide for 10 slides per class. During federation, the baseline simply concatenates the received synthetic images from other clients with its own real data for local training.

**FDD vs. Baseline:** As shown in Table 2, incorporating FDD, which replaces pixel-level generation with feature-space distillation, consistently improves MCC and leads to immediate accuracy gains. Beyond performance, FDD offers notable training efficiency by avoiding repeated feature extraction, reducing computational overhead. However,

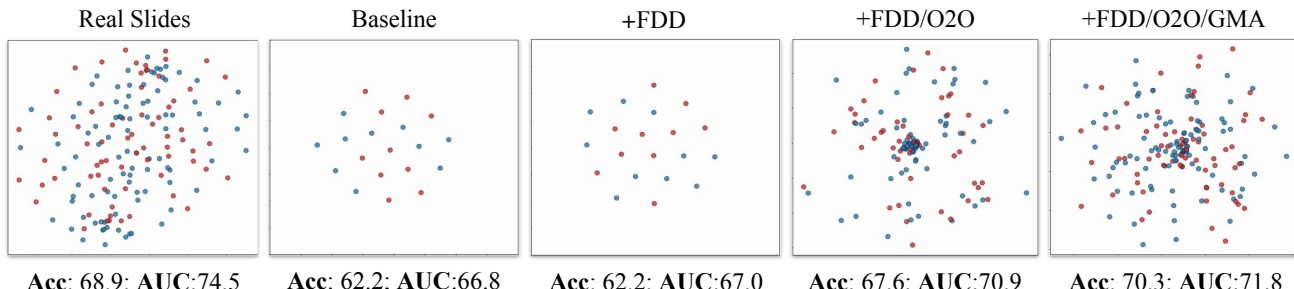

| Real Slides | Baseline | +FDD | +FDD/O2O | +FDD/O2O/GMA |
|:---:|:---:|:---:|:---:|:---:|
| **Acc**: 68.9; **AUC**:74.5 | **Acc**: 62.2; **AUC**:66.8 | **Acc**: 62.2; **AUC**:67.0 | **Acc**: 67.6; **AUC**:70.9 | **Acc**: 70.3; **AUC**:71.8 |

*Figure 2.* t-SNE visualization of patch-level feature embeddings from real slides and various ablated versions of FedHD. CAMELYON16 is used for this demonstration as it provides patch-level tumor annotations. Each point represents a patch embedding, color-coded by class (Normal vs. Tumor). We additionally report the corresponding local model performance when trained using real slides or synthetic samples to facilitate quantitative comparison.

| Baseline | Interpreted with PPR |
|:---:|:---:|

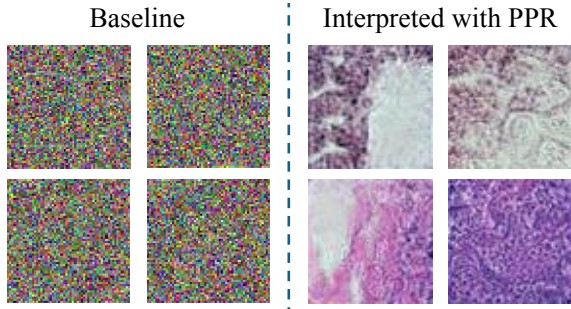

*Figure 3.* Baseline synthetic images are unrealistic. The PPR module enables realistic on-demand reconstructions without adding training overhead. The first row shows normal patches and the second row shows tumor patches.

*Table 2.* Ablation study evaluating the contribution of each proposed component in FedHD across three datasets.

| | CAM16 | | CAM17 | | IDH | |
|---|---|---|---|---|---|---|
| | Acc | MCC | Acc | MCC | Acc | MCC |
| Baseline | $88.7_{\pm4.9}$ | $75.3_{\pm12.3}$ | $77.2_{\pm6.6}$ | $52.0_{\pm13.2}$ | $80.5_{\pm4.2}$ | $47.0_{\pm10.1}$ |
| +FDD | $89.7_{\pm4.9}$ | $77.6_{\pm12.0}$ | $78.8_{\pm3.2}$ | $54.5_{\pm9.6}$ | $82.1_{\pm3.9}$ | $49.8_{\pm13.5}$ |
| +GMA | $89.4_{\pm4.9}$ | $77.1_{\pm11.9}$ | $79.9_{\pm3.8}$ | $57.1_{\pm9.5}$ | $83.4_{\pm4.1}$ | $52.4_{\pm11.6}$ |
| +CBF | $90.3_{\pm4.7}$ | $78.5_{\pm11.9}$ | $80.7_{\pm4.8}$ | $58.5_{\pm11.2}$ | $83.9_{\pm4.0}$ | $55.1_{\pm11.2}$ |
| +FDD&O2O | $90.6_{\pm4.7}$ | $79.2_{\pm11.8}$ | $81.2_{\pm4.7}$ | $60.0_{\pm10.7}$ | $84.0_{\pm4.0}$ | $54.3_{\pm11.6}$ |
| +FDD&O2O&GMA | $91.2_{\pm4.9}$ | $80.3_{\pm12.1}$ | $81.9_{\pm4.8}$ | $61.3_{\pm12.0}$ | $84.6_{\pm4.1}$ | $56.6_{\pm11.1}$ |
| +All | $\mathbf{91.2_{\pm5.0}}$ | $\mathbf{80.6_{\pm12.0}}$ | $\mathbf{82.7_{\pm4.8}}$ | $\mathbf{62.3_{\pm11.3}}$ | $\mathbf{84.8_{\pm4.1}}$ | $\mathbf{57.0_{\pm8.4}}$ |

when used alone with naive distribution modeling, it struggles to capture the full complexity of real slide features, limiting its standalone effectiveness.

**Impact of O2O:** As shown in Table 2, combining FDD with the O2O strategy consistently improves model performance. This underscores the benefit of preserving slide-wise structure during distillation, leading to more informative and generalizable embeddings. Fig. 2 further illustrates that O2O helps mitigate over-compression and retain diagnostic richness. However, due to naive distribution matching, the synthetic features still show uneven density compared to real data, motivating the need for a more expressive alignment strategy, as introduced in the GMA module.

**Impact of GMA:** Fig. 2 shows that modeling patch features as a Gaussian mixture improves alignment between synthetic and real slide distributions, resolving the uneven spread seen in naive distillation and yielding more semantically faithful embeddings. When GMA is combined with FDD and O2O, further performance gains are observed. These results confirm that higher-order alignment via mean and covariance matching is crucial for capturing morphological diversity in WSIs.

**Interpretation via Pseudo-Patch Reconstruction (PPR):** As shown in Fig. 3, we present reconstructed pseudo-patches on CAMELYON16 to qualitatively illustrate interpretability. These reconstructions support human-in-the-loop inspection and address the lack of transparency in embedding-based methods. Since synthetic features are non-invertible and contain no raw pixel data, they offer strong privacy guarantees for cross-institutional sharing. The interpretation module is locally trained and invoked only when needed, adding no overhead to training.

**Impact of CBF:** The CBF module is motivated by our empirical observation that models trained solely on distilled data, while effective, generally underperform those trained on real slides (Fig. 2 and Appendix Table 5). Notably, although models trained exclusively on distilled data remain inferior to those trained on real data, models trained using our distilled slides not only outperform prior synthetic-data-based methods (Jin et al., 2025), but also achieve performance comparable to real-data-trained models. As shown in Table 2, adding CBF gives consistent performance gain. Fig. 4 further illustrates that CBF enhances interpretability, producing more precise and diagnostically relevant heatmaps compared to naive data concatenation. In some cases (second row), it even corrects mispredictions by better localizing pathological cues.

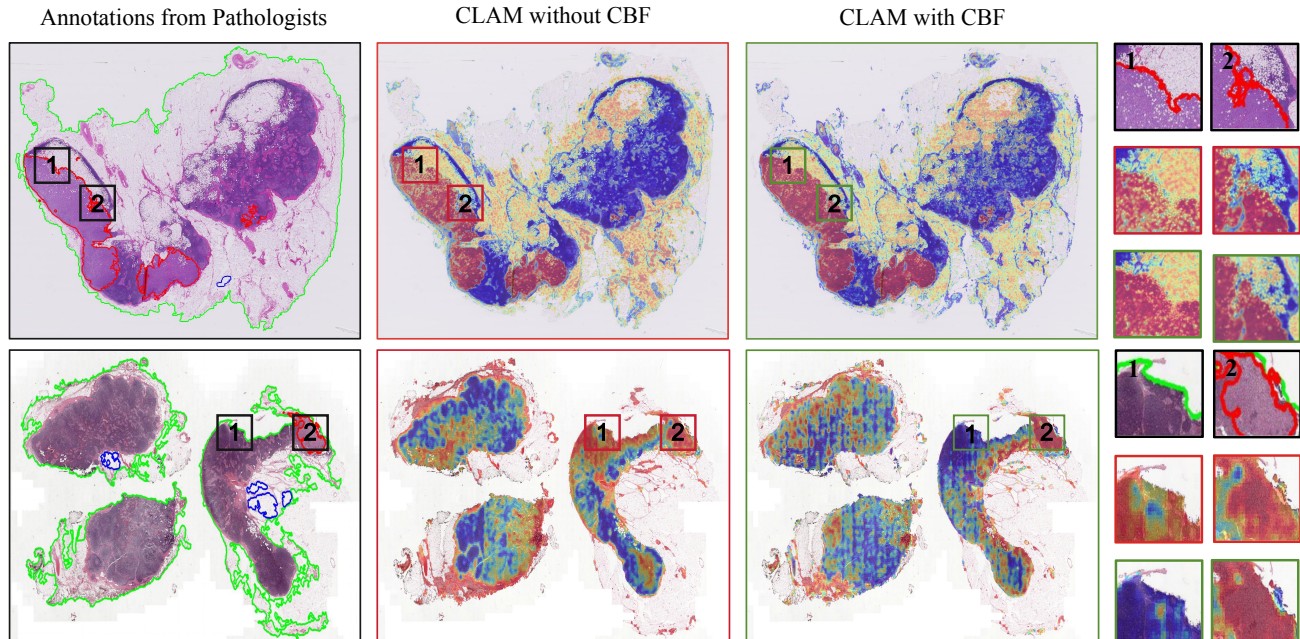

*Figure 4.* Comparison of heatmaps from a model trained with naive data concatenation versus CBF. CBF not only produces more precise and diagnostically relevant regions in successful cases (first row) but also corrects predictions by better localizing pathological cues in cases where the naive approach fails (second row).

*Table 3.* LiRA-based Membership Inference Attack (MIA) results.

|  | FedWSIDD | | FedHD (Ours) | |
| --- | --- | --- | --- | --- |
|  | Max AUC ↓ | Mean AUC ↓ | Max AUC ↓ | Mean AUC ↓ |
| CAM16 | 52.9 | 52.7 | **51.8** | **51.5** |
| CAM17 | 57.6 | 54.2 | **56.7** | **53.0** |
| IDH | 60.1 | 56.1 | **54.7** | **52.3** |

### 4.5. Privacy Analysis

The shared synthetic features in FedHD are non-invertible and contain no raw pixel information, providing strong privacy protection for cross-institutional collaboration. However, the PPR module generates image-space visualizations, which may raise concerns about potential privacy leakage. To assess this risk, we conduct a dedicated privacy analysis using the Likelihood Ratio Attack (LiRA) (Carlini et al., 2022) on the reconstructed pseudo-patch images, rather than on the feature embeddings. For each client, we split the real training slides into member (80%) and non-member (20%) sets, and generate synthetic features using only the member set. Pseudo-patches are then reconstructed via the PPR module, and cosine-distance–based likelihood ratios are computed between reconstructed samples and real images to distinguish members from non-members. As reported in Table 3, despite enabling post-hoc interpretability, FedHD consistently achieves lower AUC values than FedWSIDD, indicating reduced susceptibility to membership inference

attacks. Overall, these results demonstrate that the PPR module does not materially introduce additional privacy leakage and that FedHD preserves strong membership privacy while providing interpretable visualizations under realistic cross-client deployment conditions.

### 4.6. Limitation

The pseudo-patch reconstruction module in FedHD relies on a lightweight GAN, which may limit the visual fidelity of the reconstructed pseudo-patches. While sufficient for qualitative interpretability and human-in-the-loop inspection, more advanced generative models could further improve reconstruction quality, which we leave for future work.

## 5. Conclusions

We presented FedHD, a novel federated distillation framework tailored for WSI classification across heterogeneous institutions. By distilling semantically rich, slide-specific patch embeddings and integrating them via a curriculum-based strategy, FedHD enables privacy-preserving, architecture-agnostic collaboration. Extensive experiments show that FedHD consistently outperforms existing federated and distillation baselines, demonstrating improved generalization, scalability, and interpretability across diverse datasets and settings. Our results highlight the potential of feature-level distillation as a robust solution for federated pathology.

## Acknowledgments

This work was supported by the National Natural Science Foundation of China under Grant 62572007.

## Impact Statement

This paper presents work whose goal is to advance the field of machine learning. There are many potential societal consequences of our work, none of which we feel must be specifically highlighted here.

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

## A. Code Availability

To support reproducibility and facilitate future research, we have released the full implementation of FedHD in a public code repository. The repository contains all core components, including Feature-level Data Distillation (FDD), One-to-One Distillation Strategy (O2O), Gaussian-Mixture Feature Alignment (GMA), On-Demand Interpretation, and Curriculum-Based Federation (CBF). A detailed README file is provided to guide users through the setup and execution process. Code is available at `https://github.com/HuahuaCodes/FedHD-ICML2026`.

## B. Hyperparameter Sensitivity Analysis

We conduct a comprehensive ablation study to evaluate the sensitivity of FedHD to several key hyperparameters that influence the quality of distilled features and the stability of federated optimization. Specifically, we examine: (1) the number of synthetic patches per slide ($T$), which determines the representational capacity of the distilled dataset; (2) the number of Gaussian mixture components ($M$) used in the multi-component distribution alignment; (3) the curriculum threshold ($t_0$), which controls when cross-client synthetic data are introduced into local training; and (4) the noise robustness parameter ($q$) in the Generalized Cross-Entropy (GCE) loss, which mitigates the impact of label noise present in synthetic supervision. For each hyperparameter, we vary its value while keeping the remaining settings fixed, and report performance on all three benchmark datasets. The detailed results are presented in the following subsections.

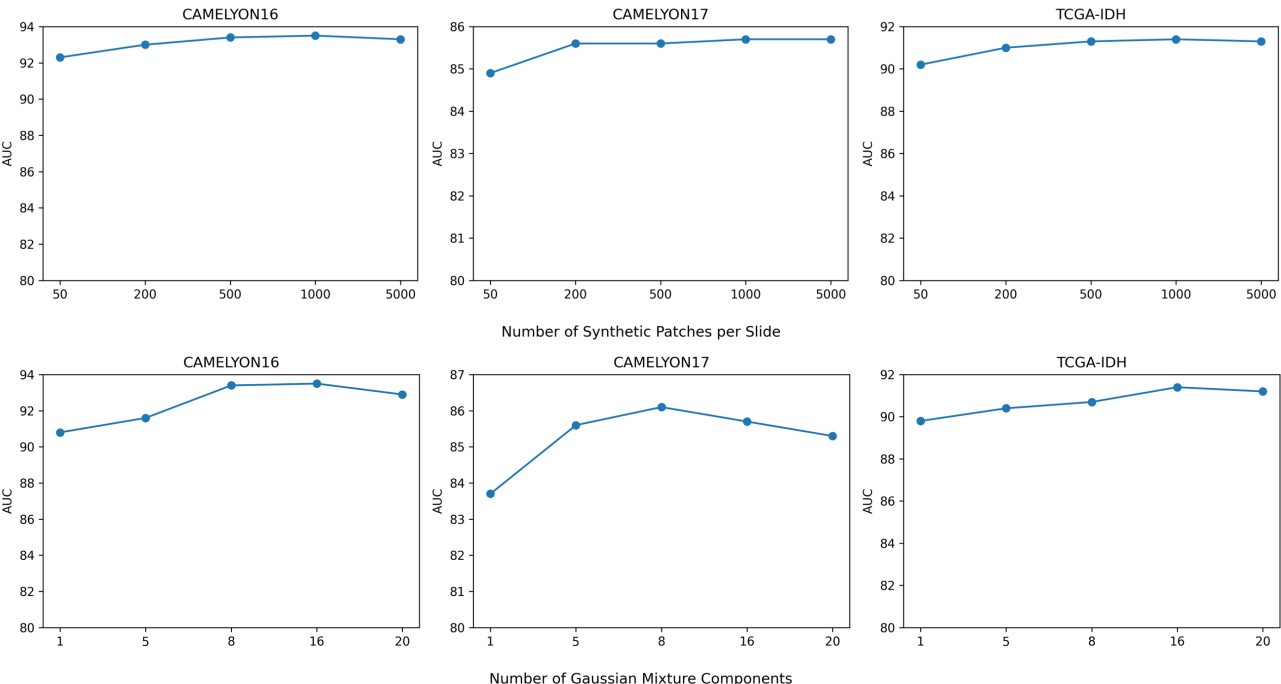

*Figure 5.* Classification performances with different number of synthetic patches per slide ($T$) (upper) and different number of Gaussian mixture components ($M$) (lower).

### B.1. Ablation Study on Different Number of Synthetic Patches per Slide ($T$)

We analyze how the number of synthetic patches per slide ($T$) affects model performance by varying it from 50 to 5000. As shown in Fig. 5 (upper), increasing $T$ improves AUC up to a point, especially between 50 and 1000. The performance tends to plateau beyond $T = 1000$, and in some cases slightly degrades, suggesting diminishing returns or potential overfitting with excessive synthetic data. This trend is consistent across all three datasets. These results indicate that a moderate value such as $T = 1000$ offers a favorable trade-off between representational richness and training efficiency.

## B.2. Ablation Study on Different Number of Gaussian Mixture Components ($M$)

To evaluate the impact of distribution modeling complexity, we vary the number of Gaussian components $M$ in the feature alignment module. As illustrated in Fig. 5 (lower), performance improves steadily from $M = 1$ to $M = 16$, confirming that multi-component alignment captures intra-slide heterogeneity more effectively than unimodal approximations. Beyond $M = 16$, performance gains are marginal or slightly decrease, potentially due to over-fragmentation of the feature space or insufficient samples per component. Based on these findings, we use $M = 16$ as a balanced choice for capturing morphological diversity without incurring additional overhead.

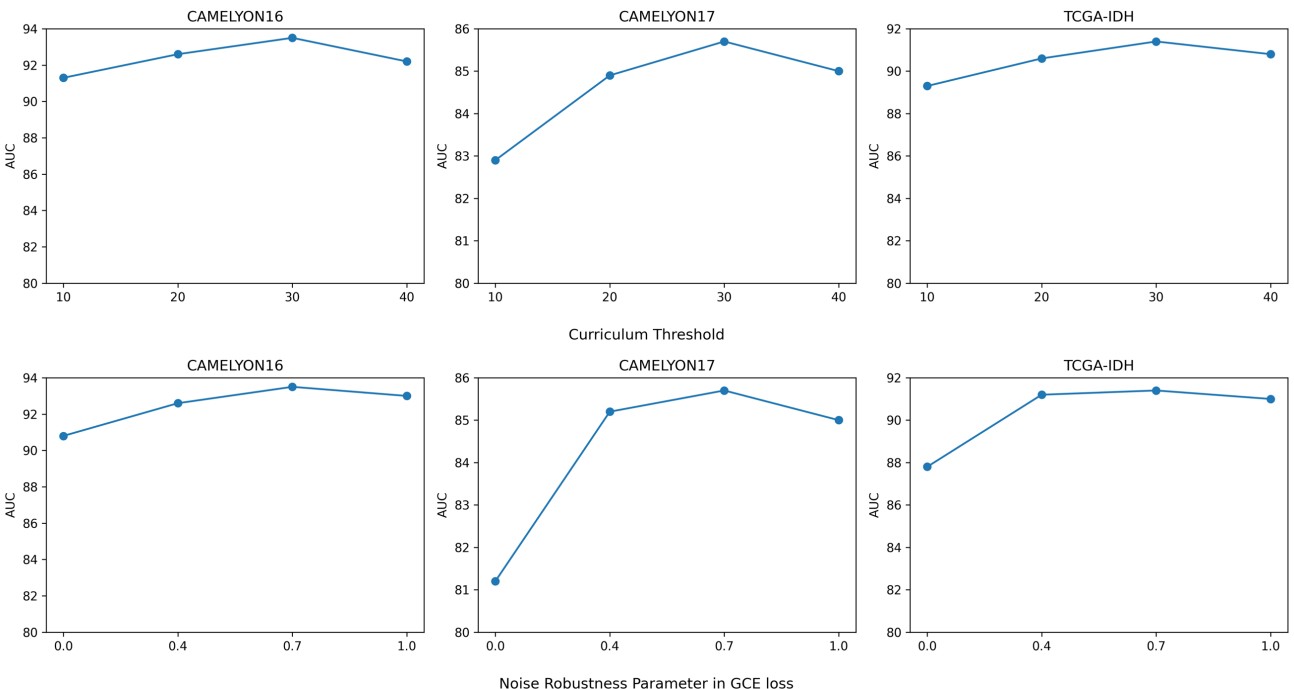

*Figure 6.* Classification performances with different curriculum threshold ($t_0$) (upper) and different noise robustness parameter in GCE loss ($q$) (lower).

## B.3. Ablation Study on Different Curriculum Threshold ($t_0$)

We investigate the effect of the curriculum threshold $t_0$, which determines when synthetic data from other clients are introduced into local training. As shown in Fig. 6 (upper), using very small values of $t_0$ (*e.g.*, $t_0 = 10$) leads to suboptimal performance due to premature exposure to cross-client synthetic data, which may contain distributional noise and hinder early model stabilization. Performance increases as $t_0$ is raised to moderate values, with $t_0 = 30$ yielding the best overall results across all datasets. However, further delaying the introduction of external synthetic samples (*e.g.*, $t_0 = 40$) slightly reduces performance, suggesting that overly conservative schedules limit the benefits of cross-site knowledge transfer. These findings highlight the importance of staged curriculum integration to balance convergence stability and collaborative learning.

## B.4. Ablation Study on Different Noise Robustness Parameter in GCE Loss ($q$)

We evaluate the robustness of FedHD under different settings of the GCE noise parameter $q$, which controls the balance between cross-entropy and MAE behavior. As illustrated in Fig. 6 (lower), small values of $q$ (*e.g.*, $q = 0.3$) overly emphasize noise-tolerant gradients and result in weaker discriminative learning, while large values approaching 1.0 diminish noise robustness and expose the model to mislabeled synthetic features. Performance consistently improves with moderate values, and $q = 0.7$ achieves the highest AUC across all datasets. These results confirm that moderately noise-robust objectives are beneficial for federated learning with distilled synthetic supervision.

## B.5. Design Rationale

Each hyperparameter in FedHD is designed to address a specific challenge in federated whole slide image (WSI) distillation. The number of synthetic patches per slide ($T$) controls the representational budget per sample: higher $T$ increases diversity but raises communication and storage costs; thus, $T$ acts as a trade-off lever between data richness and efficiency. The number of Gaussian mixture components ($M$) defines how finely the feature distribution is modeled: larger $M$ enables capturing complex morphological heterogeneity but risks over-fragmentation, especially with limited patch counts. The curriculum threshold ($t_0$) governs the timing of cross-client supervision: delaying synthetic data integration stabilizes early training, especially when incoming samples may exhibit distributional or semantic shift. Finally, the GCE noise robustness parameter ($q$) allows the model to tolerate label noise in synthetic supervision: smaller $q$ values make the loss more robust but may slow convergence, while larger $q$ improves sensitivity but increases risk of overfitting to noisy pseudo-labels. Together, these parameters form a modular design space that can be tuned to meet the needs of different federated pathology deployments.

# C. Homogeneous Model Performance

We assume all institutions adopt a unified model architecture, namely CLAM, for local MIL training, and report the results in Table 4. First, all FL methods outperform Local, reaffirming the value of cross-institutional collaboration in histopathology. Second, FedHD consistently outperforms standard FL baselines. On CAMELYON16, it achieves a weighted accuracy of 91.3%, surpassing FedMut by 3.1%. On CAMELYON17, FedHD reaches 86.0%, outperforming FedMut by 7.0%, with a similar trend observed on TCGA-IDH. Third, FedHD also surpasses state-of-the-art personalized FL methods, achieving the highest accuracy across all datasets. These results highlight the strong generalizability of FedHD. It demonstrates robust performance under both heterogeneous local models and homogeneous model architectures, making it a versatile and effective solution for federated learning in real-world applications.

*Table 4.* Homogeneous model performance comparison of FedHD against standard FL and personalized FL methods.

| Methods | | | | Standard FL | | | | | | Personalized FL | | | | | | |
|---|---|---|---|---|---|---|---|---|---|---|---|---|---|---|---|---|
| | | Local | | FedHisto | | FedImpro | | FedMut | | FedDGM | | HistoFS | | FedWSIDD | | FedHD (Ours) | |
| | | Acc | AUC | Acc | AUC | Acc | AUC | Acc | AUC | Acc | AUC | Acc | AUC | Acc | AUC | Acc | AUC |
| CAM16 | C1 | 68.9 | 74.5 | 74.3 | 79.7 | 89.2 | 87.4 | 90.5 | 90.5 | 91.9 | 90.5 | 90.5 | 88.3 | 93.2 | 91.4 | **94.6** | **97.9** |
| | C2 | 77.4 | 72.3 | 73.6 | 80.7 | 83.0 | 87.2 | 84.9 | 83.0 | 73.6 | 75.3 | 79.2 | 82.3 | 84.1 | 84.9 | **86.7** | **87.4** |
| | Avg | 72.5 | 73.6 | 74.0 | 80.1 | 86.6 | 87.3 | 88.2 | 87.4 | 84.3 | 84.2 | 85.8 | 85.8 | 89.4 | 88.7 | **91.3** | **93.5** |
| CAM17 | C1 | 70.0 | 62.3 | 75.0 | 76.8 | **90.0** | **88.9** | 85.0 | 77.4 | 75.0 | 69.9 | 80.0 | 79.2 | 85.0 | 85.8 | 80.0 | 86.1 |
| | C2 | 55.0 | 72.3 | 60.0 | 63.8 | 70.0 | 76.1 | 75.0 | 79.2 | 70.0 | 71.6 | 75.0 | 76.4 | 80.0 | 79.2 | **85.0** | **80.3** |
| | C3 | 85.0 | 64.7 | 85.0 | 80.0 | 80.0 | 77.8 | 85.0 | 84.3 | 80.0 | 81.9 | 80.0 | 80.8 | 90.0 | 85.8 | **90.0** | **93.2** |
| | C4 | 70.0 | 71.8 | 65.0 | 67.6 | 65.0 | 68.0 | 70.0 | 76.4 | 80.0 | 82.4 | 85.0 | 85.6 | 75.0 | 72.2 | **90.0** | **86.3** |
| | C5 | 70.0 | 74.0 | 80.0 | 80.8 | 75.0 | 81.3 | 80.0 | 80.8 | 80.0 | 81.0 | 75.0 | 81.0 | 80.0 | 80.8 | **85.0** | **82.6** |
| | Avg | 70.0 | 69.0 | 73.0 | 73.8 | 76.0 | 78.4 | 79.0 | 79.6 | 77.0 | 77.4 | 79.0 | 80.6 | 82.0 | 80.8 | **86.0** | **85.7** |
| IDH | C1 | 84.8 | 82.1 | 87.3 | 87.8 | 89.9 | 88.9 | 89.9 | 90.0 | 86.1 | 89.1 | 89.9 | 89.1 | 91.1 | **90.4** | **92.4** | 89.6 |
| | C2 | 68.7 | 72.5 | 68.7 | 75.4 | 75.0 | 83.1 | 75.0 | 83.6 | 87.5 | 87.5 | 81.3 | 84.3 | 81.3 | 85.9 | **87.5** | **89.1** |
| | C3 | 70.0 | 79.7 | 70.0 | 78.0 | 80.0 | 88.9 | 70.0 | 84.4 | 90.0 | 88.9 | 80.0 | 88.1 | 80.0 | 91.9 | **90.0** | **94.0** |
| | C4 | 75.0 | 74.0 | 80.0 | 88.0 | 75.0 | 81.3 | 75.0 | 84.0 | 80.0 | 82.6 | 85.0 | 85.7 | 85.0 | 89.3 | **90.0** | **97.3** |
| | C5 | 66.7 | 71.4 | 70.8 | 75.7 | 79.2 | 85.0 | 70.8 | 77.0 | 79.2 | 84.4 | 83.3 | 89.6 | **87.5** | 91.8 | 83.3 | **92.6** |
| | C6 | 70.8 | 74.4 | 73.5 | 77.8 | 73.5 | 78.8 | 76.5 | 79.3 | 82.4 | 90.4 | **85.3** | 91.4 | 79.4 | 88.9 | 85.2 | **93.1** |
| | C7 | 70.0 | 71.4 | 80.0 | 81.0 | 70.0 | 76.2 | 80.0 | 71.4 | 80.0 | 87.5 | 90.0 | **93.8** | 80.0 | 81.3 | **90.0** | 87.5 |
| | C8 | 75.0 | 76.5 | 80.0 | 76.5 | 85.0 | 78.4 | 85.0 | 86.3 | 80.0 | 80.4 | 90.0 | 86.3 | 90.0 | 88.2 | **90.0** | **90.2** |
| | Avg | 76.1 | 77.0 | 79.3 | 82.1 | 81.7 | 84.1 | 81.2 | 84.3 | 83.6 | 87.2 | 86.9 | 88.8 | 86.4 | 89.3 | **89.2** | **91.4** |

# D. More visualizations with PPR

In the main manuscript, we demonstrated that incorporating on-demand Pseudo-Patch Reconstruction (PPR) significantly improves the realism and diagnostic relevance of generated synthetic patches. As shown in Fig. 7, we provide additional qualitative examples to compare the visual quality of synthetic patches generated with and without PPR. The results clearly

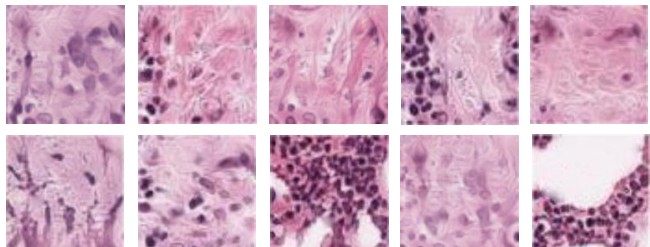

*Figure 7.* More visualizations of synthetic patches using Pseudo-Patch Reconstruction.

show that pathology guidance leads to more structured, diverse, and morphologically realistic patches that better resemble true histological patterns.

## E. Evaluation using Distilled Data

To assess the quality of distilled samples, we train a shared local MIL model (CLAM) using only the synthetic data generated by different FL+DD methods and evaluate performance on real test slides. This setup isolates the representational effectiveness of the distilled features. As shown in Table 5, FedHD achieves the highest performance across all compared methods. In the TCGA-IDH cohort, which features highly imbalanced data distributions, FedHD leads with an average accuracy of 72.1%, outperforming FedDM (Xiong et al., 2023) by 8.7 percentage points and the strongest baseline FedDGM (69.1%) by 3.0 percentage points. This consistent superiority across all three datasets demonstrates that the samples distilled by FedHD are the most informative and generalize most effectively to real-world data.

*Table 5.* Performance comparison on data distilled by different FL+DD methods.

| Methods | | DESA | | FedDM | | FedD3 | | FedDGM | | FedWSIDD | | FedHD (Ours) | |
|---|---|---|---|---|---|---|---|---|---|---|---|---|---|
| | | Acc | AUC | Acc | AUC | Acc | AUC | Acc | AUC | Acc | AUC | Acc | AUC |
| CAM16 | C1 | 67.6 | 72.9 | 67.6 | 65.6 | 62.2 | 60.5 | 68.9 | 71.5 | 67.6 | 75.3 | **70.3** | **71.8** |
| | C2 | 62.3 | 64.2 | 62.3 | 58.8 | 58.5 | 69.4 | 66.0 | 55.7 | 64.2 | 64.2 | **71.7** | **70.7** |
| | Avg | 65.4 | 69.3 | 65.4 | 62.6 | 60.7 | 64.2 | 67.7 | 64.9 | 66.2 | 70.7 | **70.9** | **71.3** |
| CAM17 | C1 | 55.0 | 61.4 | 60.0 | 61.0 | 60.0 | 55.1 | 60.0 | 61.4 | 60.0 | 66.3 | **60.0** | **70.7** |
| | C2 | 50.0 | 64.5 | 55.0 | 61.4 | 50.0 | 56.7 | 60.0 | 62.9 | 55.0 | 62.4 | **65.0** | **69.1** |
| | C3 | 75.0 | 67.7 | 70.0 | 66.3 | 65.0 | 66.4 | 80.0 | 68.2 | 75.0 | 72.0 | **80.0** | **72.8** |
| | C4 | 50.0 | 62.1 | 55.0 | 63.1 | 50.0 | 61.6 | 65.0 | 70.2 | 60.0 | 66.8 | **70.0** | **74.5** |
| | C5 | 60.0 | 62.7 | 65.0 | 60.4 | 60.0 | 64.9 | 65.0 | 67.2 | 65.0 | 70.0 | **65.0** | **72.7** |
| | Avg | 58.0 | 63.7 | 61.0 | 62.4 | 57.0 | 60.9 | 66.0 | 66.0 | 63.0 | 67.5 | **68.0** | **71.9** |
| IDH | C1 | 68.3 | 70.7 | 67.1 | 68.2 | 65.8 | 70.9 | 72.5 | 72.9 | 69.8 | 75.8 | **77.8** | **79.2** |
| | C2 | 62.5 | 61.2 | 56.3 | 62.8 | 56.3 | 56.4 | 62.5 | 61.2 | 62.5 | 68.1 | **68.8** | **71.8** |
| | C3 | 60.0 | 67.2 | 50.0 | 66.2 | 50.0 | 59.2 | 70.0 | 73.8 | 60.0 | 70.6 | **70.0** | **78.4** |
| | C4 | 60.0 | 68.9 | 60.0 | 62.1 | 55.0 | 59.3 | 70.0 | 67.7 | 60.0 | 68.9 | **70.0** | **81.6** |
| | C5 | 58.3 | 62.3 | 62.5 | 66.4 | 58.3 | 62.3 | 66.7 | 65.1 | 62.5 | 66.4 | **66.7** | **68.4** |
| | C6 | 67.6 | 67.7 | 67.6 | 61.6 | 64.7 | 63.1 | 67.6 | 64.7 | 64.7 | 68.8 | **67.6** | **70.7** |
| | C7 | 60.0 | 63.8 | 50.0 | 62.8 | 60.0 | 57.1 | 60.0 | 67.2 | 60.0 | 67.2 | **60.0** | **70.6** |
| | C8 | 70.0 | 67.7 | 65.0 | 68.4 | 60.0 | 62.9 | 70.0 | 64.6 | 70.0 | 70.9 | **75.0** | **72.5** |
| | Avg | 65.2 | 67.6 | 63.4 | 65.6 | 61.5 | 64.6 | 69.1 | 68.3 | 65.8 | 71.3 | **72.1** | **75.2** |

