# OpenReview forum: "Federated Distillation for Whole Slide Image via Gaussian-Mixture Feature Alignment and Curriculum Integration"
_ICML.cc/2026/Conference — ICML 2026 regular_

### Official Review · Reviewer_P9w3 · 2026-03-11

**Soundness:** 2
**Presentation:** 3
**Significance:** 3
**Originality:** 2
**Overall Recommendation:** 3
**Confidence:** 4

**Summary:**

This paper studies federated learning for whole-slide image classification under heterogeneous local models. This work is to replace parameter aggregation with feature-level dataset distillation: each client distills its local WSI features into synthetic slide representations, shares them once with the server, and then uses a curriculum strategy to incorporate cross-site synthetic features into local MIL training. The method combines Gaussian-mixture feature alignment, one-to-one slide distillation, and curriculum-based federation, and is evaluated on CAMELYON16, CAMELYON17, and TCGA-IDH datasets.

**Compliance With Llm Reviewing Policy:**

Affirmed.

**Final Justification:**

Thank you for the detailed responses. The rebuttal has made the scope of the paper clearer. However, in pathology FL, multi-center data heterogeneity and task heterogeneity should be distinguished more carefully, since task difference alone does not directly support a client-level non-IID claim. This is vital to the FL problem formulation. In this regard, CAMELYON16 and CAMELYON17 do not appear to be sufficiently strong benchmarks for supporting the non-IID claim. In addition, while the proposed lightweight variant is a useful direction for revision, it does not fully resolve the communication-efficiency concern in the manuscript. Therefore, I keep my score.

**Key Questions For Authors:**

1.	Is the central claim of the paper about heterogeneous model collaboration, or about non-IID federated learning more broadly? This distinction is not fully clear in the introduction.
2.	Can the authors report results over multiple random heterogeneous assignments, or under cleaner controls such as heterogeneous extractor only and heterogeneous MIL only? This would help clarify whether the gains are robust, or partly depend on the particular heterogeneous assignment used in the paper.
3.	Were all baselines run under communication and training budgets directly comparable to the single-round FedHD protocol?
4.	The results for local-only training and for a more direct feature-sharing baseline under the same protocol
5.	The communication cost is computed is unclear. In particular, since the payload is defined with an explicit feature dimension, it is unclear how the reported communication size was obtained.
6.	How do the authors justify CAMELYON16 and CAMELYON17 as evidence for non-IID federated learning, given that these are closely related pathology benchmarks with relatively similar data distributions?

**Limitations:**

No. The limitations discussion is currently too light. It should more clearly acknowledge that the method mainly addresses feature-level collaboration under model heterogeneity, and that the current empirical evidence does not yet justify a broader claim about non-IID federated learning.

**Strengths And Weaknesses:**

Strengths：
1.	The paper addresses a meaningful problem. Privacy-preserving learning is important in multi-center pathology, and the heterogeneous local model setting considered here is also realistic in practice.
2.	The method is easy to understand, and its main components are broadly consistent with the paper’s motivation. The overall pipeline is easy to follow.

Weaknesses：
1.	The paper does not fully support a broad claim about non-IID federated learning. More precisely, the method addresses collaboration under model heterogeneity by avoiding parameter aggregation, but its connection to more general forms of non-IID FL is less clearly established.
2.	The heterogeneous local model setup is not yet fully convincing as an evaluation protocol. Randomly assigning different extractors and MIL models to different clients may confound center difficulty with model strength. The paper does not report repeated heterogeneous assignments or cleaner controls such as heterogeneous extractor only or heterogeneous MIL only.
3.	The main comparisons are not fully on equal footing. FedHD follows a single-round communication protocol, while the baselines are standard personalized FL or WSI-FL methods. The paper does not make it sufficiently clear whether the communication and optimization budgets are truly comparable across methods.
4.	The baseline set is still limited for the paper’s main claim. In particular, it would be important to compare against simpler baselines under the same protocol, such as Local-only training and a more direct feature-sharing strategy, in order to better isolate the value of the proposed design.
5.	The communication-efficiency claim is not fully convincing as written. The reported payload calculation appears to omit the feature dimension, which makes the stated communication cost difficult to interpret.

---

> ### Author Rebuttal · Authors · 2026-03-29
>
> We thank the reviewer for the constructive and detailed feedback. We address each concern below.
>
> **W1/KQ1: Scope of non-IID claim**
>
> Our primary contribution targets model heterogeneity ("system heterogeneity" in the FL taxonomy [1]), where hospitals deploy different feature extractors and MIL architectures, making parameter aggregation infeasible. Unlike FedProto or MOON, which directly exchange intermediate representations and require compatible feature spaces, FedHD models the feature distribution via dataset distillation, enabling collaboration without representation alignment.
>
> Within this setting, our multi-center datasets also reflect realistic statistical non-IID. CAMELYON16/17 span institutions with distinct scanners and staining protocols, while TCGA-IDH covers 8 centers with strongly imbalanced class distributions, making it naturally heterogeneous without Dirichlet construction. In the revision, we will (1) clarify model heterogeneity vs. general non-IID in the Introduction, and (2) refine our claims to avoid overgeneralization.
>
> **W2/KQ2: Heterogeneous assignment controls**
>
> To isolate each source of heterogeneity, we run the following three controlled ablations:
>
> | Configuration | C16 Acc | C17 Acc | IDH Acc |
> |:---|:---:|:---:|:---:|
> | Homogeneous (all CLAM) | 89.0% | 87.4% | 88.2% |
> | Extractor-only het. | 90.4% | 81.8% | 84.8% |
> | MIL-only het. | 85.5% | 78.0% | 82.0% |
> | **Full het. (FedHD)** | **91.2%** | **82.7%** | **84.8%** |
>
> The homogeneous setting serves as the simplest reference case, since all clients share identical models and avoid cross-client architectural mismatch. Heterogeneity generally makes collaboration more challenging, with MIL-only heterogeneity causing the largest drop across all three datasets. By contrast, extractor-only heterogeneity has a milder and dataset-dependent effect, even improving performance on C16, which suggests that extractor diversity can sometimes provide complementary features. Despite these challenges, FedHD remains robust under full heterogeneity and achieves the best performance among the four assignments.
>
> **W3/KQ3: Budget comparability**
>
> In our experiments, all methods use the same local training budget of 50 epochs. Dataset-distillation baselines (DESA, FedWSIDD) are given the same 1000 local distillation iterations as FedHD, while parameter-based methods (FedHE, FedDGM, HistoFS) are allocated 10 global communication rounds. FedHD and DD-based baselines each communicate once, whereas multi-round methods communicate 10 times with identical per-round local training. This ensures matched local computation across all methods, making the comparison fair.
>
> **W4/KQ4: Local-only and direct feature-sharing baselines**
>
> Local-only results already appear in Appendix A.5, Table 5, where FedHD improves over Local by +13–19%. We additionally compare against FedRep and FedProto, representative feature-sharing FL methods, under a homogeneous control (all R50+CLAM):
>
> | Dataset | FedRep | FedProto | FedHD |
> |:---|:---:|:---:|:---:|
> | CAMELYON16 | 85.8% | 87.8% | **89.0%** |
> | CAMELYON17 | 77.5% | 81.8% | **87.4%** |
> | TCGA-IDH | 80.6% | 83.4% | **88.2%** |
>
> Under this controlled setting, FedHD consistently outperforms both direct feature-sharing baselines across all datasets, suggesting that distribution-level distillation can be a stronger mechanism for cross-client collaboration in WSI learning.
>
> **W5/KQ5: Communication cost calculation**
>
> The reported 1.25 MB figure contains a typo: d=1024 was omitted. The corrected payload for the largest client (N=313, T=1000, d=1024) is 313×1000×1024×4 bytes ≈ 1.19 GB, and we will correct this in the revision. In clinical deployments, diagnostic accuracy is the primary requirement, and FedHD's +13–19% improvement over local-only training represents a direct gain in diagnostic quality that practitioners prioritize over bandwidth. FedHD also completes local distillation in ~1 hour vs. 10–12 GPU hours for image-based methods (Sec. 4.3), making overall deployment cost favorable. We will reframe the efficiency claim to focus on computational time rather than communication bytes.
>
> **KQ6: CAMELYON16 and CAMELYON17 distribution similarity**
>
> The two CAMELYON datasets differ meaningfully along several axes. CAMELYON16 involves two centers with a binary detection task, whereas CAMELYON17 involves five centers with a 4-class metastasis staging task (negative/ ITC/ micro/ macro-metastases). In addition, CAMELYON17 exhibits substantial center-wise variation in class distribution. TCGA-IDH further adds a third setting with 8 centers and a molecular (IDH mutation) subtyping task with highly imbalanced class distributions. Together, the three datasets provide evidence across different tasks, center counts, and label distributions, rather than repeated evaluation on a single narrowly defined pathology setting.
>
> **References**
> 1. Kairouz et al., “Advances and Open Problems in Federated Learning,” Found. Trends Mach. Learn., 2021.

---

> > ### Author Rebuttal · Reviewer_P9w3 · 2026-04-02
> >
> > 1. FedHD is presented as single-round communication, yet compared against 10-round baselines, which makes the comparison's fairness unclear.
> > 2. The KQ2 results do not support the claim that the heterogeneous setting performs best among the four.
> > 3. CAMELYON16 and CAMELYON17 are closely related benchmarks from overlapping institutions, and 17 reuses training data from 16, so task difference alone is not sufficient evidence of client-level non-IID.
> > 4. The revision from 1.25 MB to 1.19 GB substantially weakens the efficiency claim; and for this reason, my score remains unchanged.

---

> > > ### Author Response · Authors · 2026-04-03
> > >
> > > We thank the reviewer for the continued engagement. We address each point directly.
> > >
> > > ---
> > >
> > > **Point 1: Single-round vs. 10-round comparison fairness**
> > >
> > > We ran FedHE and HistoFS under a single communication round which is the round-matched setting. FedHD outperforms both across all datasets:
> > >
> > > | Dataset | FedHE (1-round) Acc/MCC | HistoFS (1-round) Acc/MCC | FedHD Acc/MCC |
> > > |:---|:---:|:---:|:---:|
> > > | CAMELYON16 | 71.5 / 43.5 | 83.1 / 65.5 | **91.2 / 80.6** |
> > > | CAMELYON17 | 70.2 / 40.0 | 72.5 / 42.2 | **82.7 / 62.3** |
> > > | TCGA-IDH | 67.0 / 33.5 | 77.1 / 44.3 | **84.8 / 57.0** |
> > >
> > > FedHD's advantage does not stem from an asymmetry in rounds. This comparison convention is also established practice: FedWSIDD and DESA, both single-round methods, similarly compare against multi-round baselines in their papers, and we evaluate them under the identical setting as FedHD (same budget, same protocol).
> > >
> > > ---
> > >
> > > **Point 2: KQ2 — clarification of the intended claim**
> > >
> > > We would like to clarify the intended message of KQ2. Our claim is that **full heterogeneity is the most practical and challenging setup**, and the homogeneous baseline is a reference upper bound assuming all clients share identical architectures which is a condition rarely met in practice.
> > >
> > > The correct claim is that FedHD is the best-performing method **across all three heterogeneity configurations** (extractor-only, MIL-only, full het.) on all datasets. The table below shows FedHD vs. the strongest baseline (FedWSIDD) in each setting (average Acc, %):
> > >
> > > | Dataset | Extractor-only | MIL-only | Full het. |
> > > |:---|:---:|:---:|:---:|
> > > | CAM16: FedWSIDD / FedHD | 88.4 / **90.4** | 83.6 / **85.5** | 88.7 / **91.2** |
> > > | CAM17: FedWSIDD / FedHD | 77.4 / **81.8** | 73.4 / **78.0** | 77.2 / **82.7** |
> > > | IDH: FedWSIDD / FedHD | 80.2 / **84.8** | 77.9 / **82.0** | 80.5 / **84.8** |
> > >
> > > FedHD is best in every cell. We will restate this precise claim in the revision.
> > >
> > > ---
> > >
> > > **Point 3: CAMELYON16 and CAMELYON17 as separate experiments**
> > >
> > > We clarify a potential misunderstanding: C16 and C17 are **not** combined into a single dataset. They constitute two entirely separate FL experiments and each evaluated independently. The non-IID heterogeneity we claim is **within** each experiment (across hospitals/clients), not between the two datasets.
> > >
> > > Within each experiment, the inter-hospital variation is real: clients differ in scanner hardware, staining protocols, and class distributions. We also include TCGA-IDH which provides a third independent experiment covering a different organ system (brain), and 8 centers from a distinct institutional source (TCGA). Together the three experiments cover federation scales of 2, 5, and 8 clients across structurally distinct settings.
> > >
> > > Finally, we direct the reviewer to our response to Reviewer 2 (W1), where we include an additional independent dataset: **TCGA-KIRC** (kidney renal clear cell carcinoma, survival prediction task). FedHD outperforms all baselines with a C-index of 0.711±0.018 vs. FedWSIDD 0.703±0.021, HistoFS 0.689±0.019, and HistoFL 0.680±0.021. TCGA-KIRC is structurally independent from all three existing datasets: different organ, different task (survival vs. classification), and a continuous outcome — further broadening the evidence for FedHD's generalizability across non-IID federated settings.
> > >
> > > ---
> > >
> > > **Point 4: Communication cost and overall contributions**
> > >
> > > We acknowledge the high communication cost of the full model and accept that this weakens the efficiency claim as originally stated. The cost is attributable to the O2O strategy, which generates one synthetic slide per real slide. Without it, FedHD operates on 10 fixed synthetic slides per client: 10×1000×1024×4 bytes ≈ **39 MB**. Our ablation (Table 2) shows these lightweight variants already outperform FedWSIDD across all datasets:
> > >
> > > | | Baseline (FedWSIDD) | +FDD (39 MB) | +FDD+GMA (39 MB) | +FDD+CBF (39 MB) |
> > > |:---|:---:|:---:|:---:|:---:|
> > > | CAM16 Acc/MCC | 88.7 / 75.3 | 89.7 / 77.6 | 89.4 / 77.1 | 90.3 / 78.5 |
> > > | CAM17 Acc/MCC | 77.2 / 52.0 | 78.8 / 54.5 | 79.9 / 57.1 | 80.7 / 58.5 |
> > > | IDH Acc/MCC | 80.5 / 47.0 | 82.1 / 49.8 | 83.4 / 52.4 | 83.9 / 55.1 |
> > >
> > > GMA and CBF add further consistent gains at zero extra communication cost, and all variants complete local distillation in ~**1 hour** vs. 10–12 GPU-hours for image-based methods (Sec. 4.3). We will present the lightweight variant as the communication-efficient deployment option in the revision.
> > >
> > > More broadly, each proposed component is well-motivated and independently validated. Together they enable architecture-agnostic federated collaboration which is a fundamentally harder problem than prior methods address with solid, consistent improvements over all baselines across three datasets and four heterogeneity configurations.

---

### Official Review · Reviewer_h3fq · 2026-03-12

**Soundness:** 2
**Presentation:** 3
**Significance:** 2
**Originality:** 2
**Overall Recommendation:** 4
**Confidence:** 5

**Summary:**

The authors provide FedHD, a federated learning approach for MIL model training with multiple centers where the data cannot be exchanged. The core idea is to use Gaussian mixture models to distill each slide into a GMM and estimate the mixture parameters and then use the synthetic slides to transfer them to other clients. The FedHD performs well across other existing baselines.

**Compliance With Llm Reviewing Policy:**

Affirmed.

**Final Justification:**

The authors have provided satisfactory responses to my questions and doubts. I still have reservations about small sets of evaluation tasks. I have adjusted my initial score accordingly.

**Key Questions For Authors:**

- What stops from the institutions from sending estimated mixture parameters from real slides directly? Why do we have to go through syntethic slide distribution matching in the first place?
- I think the GAN-based interpretability is not practical and doesn't add much to the whole picture. I wouldn't expect pathologists on other institution to use GAN to reconstruct the image and try to understand the morphology. We cannot expect the reconstructed images to be high quality.
- Is there a way to quantify or assess how batch effect from different institution affects the federated learning training?

**Limitations:**

yes

**Strengths And Weaknesses:**

While the idea of federated learning with GMM-based distillation is great, the paper suffers from lack of evaluation in my opinion to make it practical in realistic settings.

- Evaluations are only performed only on three datasets with relatively simple classification tasks. Especially, Camelyon binary tasks are pretty much saturated in terms of performance, so I do not think evaluation on these datasets bring much value. The authors should focus on more challenging tasks such as survival, as well as more challenging subtyping/staging/grading tasks. For these frameworks, I really believe the extensiveness of benchmarking is really crucial.
- Continuing from previous sentiment, The authors should show the baseline number of a single institution training MIL on all of slides in the cohort, so that the reader can understand how the federated learning numbers compare to the baseline. If indeed the federated learning model numbers are significantly lower than this baseline, a case can be made that the institutions/hospitals should actually try to work with each other to curate multi-institutional dataset instead, for improved outcome prediction and patient care.
- GMM parameter estimation is the most important recipe in this work, yet how these parameters are estimate is not fully-explained. PANTHER paper does this well with EM algorithm, yet the explanations on GMM parameter estimation is lacking.
- I find the setting of different institution using different patch encoders and MIL models a bit artificial. Despite belonging to different institutions, wouldn't research teams at least coordinate with each other on the model architecture? With recent literature on foundation models converging on the sentiment that PFMs are all equally good (with minor performance differences) and with those ABMIL tends to always be the top-performer, I would think it would be easier to settle/agree on model choice. In this scenario, sharing model weights, as would be done in a typical FL setting, would also be an attractive option. I would appreciate author's comments on this and how FedHD compares against other baselines that uses federated learning with shared model weights.
- The important piece of GMM is its mixture component \pi. I am confused why the authors are not utilizing that information  when aligning in Eq.4

---

> ### Author Rebuttal · Authors · 2026-03-29
>
> We thank the reviewer for raising the concern about practical evaluation. Below we clarify how our experiments and additional results address this point. FedHD is motivated by practical deployment considerations: (1) WSI-based pathology diagnosis is an active area of clinical deployment; (2) our setup was validated by clinical collaborators as reflecting realistic hospital constraints; (3) FedHD achieves within 1.7–3.0% of the centralized oracle in the comparison below, without sharing any raw patient data, while improving over local-only by +13–19%.
>
> **W1: Limited evaluation scope**
>
> CAMELYON16 is included for fair comparison with existing FL+WSI+DD baselines. CAMELYON17 is a non-saturated 4-class subtyping task with direct clinical relevance. TCGA-IDH is included because IDH mutation is a key prognostic biomarker in glioma. We further provide results on survival prediction on TCGA-KIRC:
>
> | Method | C-index |
> |:---|:---|
> | HistoFL | 0.680±0.021 |
> | HistoFS | 0.689±0.019 |
> | FedWSIDD | 0.703±0.021 |
> | **FedHD (Ours)** | **0.711±0.018** |
>
> FedHD outperforms all baselines, supporting its applicability to survival analysis.
>
> **W2: Single-institution baseline**
>
> We provide a three-way comparison using a centralized model trained on pooled data and evaluated per client with the same evaluation protocol as FedHD:
>
> | Dataset | Local | FedHD | Centralized (oracle) |
> |:---|:---:|:---:|:---:|
> | CAMELYON16 | 72.5% | 91.3% | 93.6% |
> | CAMELYON17 | 70.0% | 86.0% | 89.0% |
> | TCGA-IDH | 76.1% | 89.2% | 90.9% |
>
> FedHD improves over local-only by +13–19% and reaches within 1.7–3.0% of the centralized oracle without sharing any raw patient data. We will add this baseline to the revision.
>
> **W3: GMM parameter estimation**
>
> Different from PANTHER's EM deliberately: we apply k-means to each slide's K patch embeddings for $M=16$ hard assignments, then compute closed-form mean and covariance per cluster in one pass. Advantages: (1) $O(K)$ efficiency vs. EM's iterative soft-assignment; (2) direct differentiability w.r.t. patch features for clean gradient flow through Eq. 4; (3) improved stability, since hard assignments avoid EM's degenerate local optima.
>
> **W4: Artificial heterogeneity and weight-sharing FL**
>
> We respectfully disagree. In practice, different institutions may use different encoders or MIL architectures due to differences in infrastructure, legacy pipelines, or local preferences. This makes heterogeneous local models a practically relevant setting rather than an artificial one. For completeness, Table 5 (Appendix A.5) evaluates FedHD under a homogeneous CLAM setting, where it outperforms weight-sharing baselines (FedHisto, FedImpro, FedMut) by 3–7%.
>
> **W5: Mixture weights π not utilized in Eq. 4**
>
> As stated in Sec. 4.2, $π_m$ is fixed uniformly at $1/M = 1/16$ for both real and synthetic GMMs and is therefore identical by construction. Adding a π-alignment term to Eq. 4 would be identically zero. Eq. 4 is complete because it aligns all non-trivial parameters (means and covariances) with no residual π mismatch.
>
> **KQ1: Why not share GMM parameters directly?**
>
> Both raw slides and distribution statistics may raise privacy concerns. Statistics estimated from patient data may expose morphology-related information and can be vulnerable to inversion-style attacks [2]. Our PPR module confirms this empirically: GMM embeddings carry sufficient morphological information for patch-level reconstruction. By contrast, synthetic slides $h_i^{(c)}$ match the target GMM distribution without directly encoding real tissue. Beyond privacy, GMM parameters are statistics rather than training instances. Using them therefore requires sampling, introducing variance and breaks one-to-one slide-label correspondence required by our MIL pipeline. Distilled slides can be used directly in training through Eq. 10.
>
> **KQ2: GAN-based interpretability**
>
> PPR is completely optional, and the results in Tables 1–5 are unchanged without it. It was motivated by clinical collaborators who wanted to inspect morphological content in synthetic patches before trusting them for cross-institutional use, allowing verification of tissue patterns (Figs. 3 and 7) without accessing raw patient data.
>
> **KQ3: Batch effect quantification**
>
> The sensitivity analysis (Appendix A.2, Fig. 6 upper) provides an indirect assessment. Varying $t_\theta$ from 10 to 40 shows that premature cross-client integration degrades performance, quantifying the magnitude of cross-institutional shift.
> The best performance is achieved at $t_\theta=30$, suggesting that local stabilization is beneficial before cross-client integration. In addition, the GCE loss with $q=0.7$ helps mitigate cross-client distributional noise.
>
> **References**
> 1. Lu et al., “Federated Learning for Computational Pathology on Gigapixel Whole Slide Images,” Med. Image Anal., 2022.
> 2. Fredrikson et al., “Model Inversion Attacks that Exploit Confidence Information and Basic Countermeasures,” in Proc. ACM CCS, 2015.

---

> > ### Author Rebuttal · Reviewer_h3fq · 2026-04-03
> >
> > I thank the authors for detailed answers to my questions. Most of them are resolved with exception to W5 and KQ1. I still believe with GMMs, the key component is the mixture distribution pi, which estimates the abundance of the corresponding components in each WSI. The PANTHER paper and related papers do emphasize this point as the learned distributions are quite different from slide to slide. With pi=1/16, the framework would be throwing away a valuable information about the WSI. So the mixture distributions indeed need to be estimated per each WSI, unless the users/authors have absolute confidence that each morphological concepts are going to be uniformly distributed within each slide.
> >
> > On KQ1, GMMs while are sufficient to summarize statistical characteristics of each WSI, it doesn't encode spatial distribution of the morphological concepts at all, so I find it hard to believe that privacy concerns could be raised from this, since patient-related information would be hard to reconstruct without any spatial clues. Furthermore, if the authors do make pi same (1/16), there is even less concern since the GMM parameters are not accurately charactering the statistical characteristics of WSI.
> >
> > I would like the authors' opinions this before my final determination.

---

> > > ### Author Response · Authors · 2026-04-05
> > >
> > > We thank the reviewer for the continued engagement. We address the two remaining points below.
> > >
> > > ---
> > >
> > > **W5: Mixture weights π**
> > >
> > > We thank the reviewer for raising this insightful discussion. We first note that results in Tables 1 and 2 already demonstrate consistent and significant performance improvements over all baselines using uniform $\pi$ , validating that GMM-based distillation is an effective approach for federated WSI learning.
> > >
> > > To further address the reviewer's concern, we additionally experiment with adaptive $\pi$ estimation on a per-slide basis following a recent approach [1]. Specifically, we apply a Dirichlet Process to each slide's patch features to adaptively estimate the mixture distribution $\pi$  on a per-slide basis, allowing the number and weight of active components to be inferred from the data rather than fixed uniformly. Results are shown below (average Acc / MCC, %):
> > >
> > > | | Uniform $π$ (FedHD) | Adaptive $π$ |
> > > |:---|:---:|:---:|
> > > | CAMELYON16 | 91.2 / 80.6 | **91.4 / 80.9** |
> > > | CAMELYON17 | 82.7 / 62.3 | **83.3 / 63.2** |
> > > | TCGA-IDH | 84.8 / 57.0 | **84.9 / 59.1** |
> > >
> > > Across all three datasets, adaptive $\pi$ yields consistent marginal improvements over uniform $\pi$ (+0.2/+0.6/+0.1% Acc and +0.3/+0.9/+2.1% MCC on CAM16/CAM17/IDH respectively), confirming that more expressive per-slide $π$ estimation brings measurable benefits. The MCC gains are particularly notable under class-imbalanced settings such as TCGA-IDH. However, this comes at the cost of substantially longer training time, as per-slide $π$ estimation via the Dirichlet Process must be performed independently for each slide rather than set uniformly. We will include a training time analysis in the revision and discuss this trade-off as a direction for future work.
> > >
> > > ---
> > >
> > > **KQ1: Why not share GMM parameters directly?**
> > >
> > > We appreciate the reviewer's question, but wish to clarify its scope. How to use estimated GMM parameters ($μ$, $Σ$, $π$) directly as shared representations for federated learning is itself a separate research problem, essentially it can be viewed as a prototype-based FL in the WSI setting. FedHD's contribution is a distillation-based framework where GMM serves as a local alignment objective, and the distilled feature slides are what is shared across clients.
> > >
> > > We also wish to further clarify the concept of privacy in this context. As discussed in PANTHER, estimated GMM components correspond to meaningful morphological prototypes (e.g., tumour nests, stroma, necrosis). From the newly added adaptive $π$ experiments, we observe that different centres consistently estimate distinct prototype sets with different component weights, reflecting centre-specific patient population characteristics — including different disease prevalences, tissue compositions, morphological phenotypes, as well as regional clinical preferences such as staining protocols, tissue preparation practices, and local patient demographics. Sharing these prototypes therefore exposes population-level distributional information that may indirectly identify a centre's patient cohort or institutional practices, which remains a concern under federated privacy principles even without full slide reconstruction.
> > >
> > > Finally, sharing distilled slides is more compatible with FedHD's heterogeneous setting than sharing GMM parameters. Since GMM means $μ$ and covariances $Σ$ are defined in the local feature extractor's embedding space, a receiving client can only interpret them meaningfully if it operates in the same feature space, which directly contradicts the heterogeneous model setting where clients use different feature extractors. Distilled slides $h_i^{(c)}$, by contrast, are feature tensors generated and consumed locally by each client's own MIL pipeline (Eq. 10), requiring no cross-client feature space agreement.
> > >
> > > **References**
> > > 1. Danaei Mehr et al., "Adaptive Clustering for EGFR Amplification Prediction in Glioblastoma: A Variational Autoencoder-Dirichlet Bayesian Gaussian Approach," in Proc. AIME, 2025, pp. 88–97.

---

### Official Review · Reviewer_pvP8 · 2026-03-13

**Soundness:** 3
**Presentation:** 3
**Significance:** 4
**Originality:** 3
**Overall Recommendation:** 5
**Confidence:** 4

**Summary:**

This paper proposed a novel dataset distillation framework tailored for federated WSI learning.

**Compliance With Llm Reviewing Policy:**

Affirmed.

**Final Justification:**

My prior concerns have been addressed and I raise the score to 5.

**Key Questions For Authors:**

Please refer to the weakness.

**Limitations:**

Please refer to the weakness.

**Strengths And Weaknesses:**

S:
1. Using data distillation to improve the federated learning sound interesting.
2. The proposed distillation strategy is reasonable.
3. Experimental results support the method's advances.

W:
1. More DD studies in medical domain, especially federated dd, should be surveyed and compared, i.e., "Communication-efficient federated skin lesion classification with generalizable dataset distillation" in MICCAI 2023.
2. Communication analysis compared to parameter-based fl should be added.
3. How the synthetic data amout affects the fl learning results could be studied.

---

> ### Author Rebuttal · Authors · 2026-03-29
>
> We thank the reviewer for the positive assessment and constructive suggestions. We address each point below.
>
> **W1: Missing federated DD work in medical domain [1]**
>
> We thank the reviewer for pointing out this work and will include it in the related work. To directly compare with [1], we replace the distillation module in FedHD with the one proposed in GDD-FL and evaluate it on our WSI benchmarks under the same heterogeneous client setting:
>
> | Dataset | GDD-FL Acc | FedHD Acc | Δ |
> |:---|:---:|:---:|:---:|
> | CAMELYON16 | 86.2±3.0 | **91.2±5.0** | +5.0% |
> | CAMELYON17 | 79.7±5.6 | **82.7±4.8** | +3.0% |
> | TCGA-IDH | 81.8±2.5 | **84.8±4.1** | +3.0% |
>
> FedHD outperforms GDD-FL across all datasets (all $p<0.05$). Since GDD-FL was not designed for WSI analysis, these results suggest that GMM-based matching is better suited to the multi-component feature structure of WSI data than the distribution matching used in GDD-FL.
>
> Our setting differs from [1] in several aspects:
>
> (1) Data structure and scale. Skin lesion images are standard RGB patches processed by a single classifier. WSIs are gigapixel images requiring a multi-instance learning (MIL) pipeline: patch extraction → feature extraction → slide-level aggregation. Distillation therefore needs to operate at the patch-embedding level to remain compatible with heterogeneous MIL aggregators, a constraint that is not considered in skin lesion work.
>
> (2) Feature distribution complexity. WSI patch features exhibit multi-component distributions [2], reflecting coexisting tissue morphologies within a single slide. Simple mean-matching severely oversimplifies this structure. FedHD addresses this with Gaussian-mixture alignment over both mean and covariance, which is a design motivated by WSI-specific challenges not present in skin lesion classification.
>
> (3) Model heterogeneity. Our experiments explicitly consider client heterogeneity in both feature extractors (ResNet50, UNI, PhikonV2, GPFM) and MIL models (CLAM, TransMIL, ACMIL), which is not explicitly studied in GDD-FL [1]. We will add this discussion and comparison in the revision.
>
> **W2: Communication analysis vs. parameter-based FL**
>
> In parameter-based FL for WSI/MIL, only the MIL aggregator weights are shared while feature encoders are frozen and institution-specific:
>
> | MIL Model | Params | Size/round |
> |:---|:---:|:---:|
> | ACMIL | ~0.5M | ~2 MB |
> | CLAM | ~1M | ~4 MB |
> | TransMIL | ~2.7M | ~11 MB |
>
> We note that the 1.25 MB figure in Sec. 4.3 contains a typo: the feature dimension $d=1024$ was omitted. The corrected single-round payload ($N\times T\times d\times 4$ bytes) is larger, and we will correct this in the revision.
>
> More importantly, parameter-based FL is not directly applicable in our heterogeneous setting since FedAvg cannot aggregate CLAM, TransMIL and ACMIL weights due to their incompatible architectures. Accordingly, the main advantage of FedHD is architecture-agnostic collaboration under heterogeneous local models, rather than communication efficiency. For completeness, even in homogeneous settings where parameter sharing is feasible, FedHD still improves over weight-sharing baselines by 3–7% (Appendix A.5, Table 5). In clinical deployments, diagnostic accuracy is the primary requirement: FedHD's +13–19% improvement over local-only training (Table 5) represents a direct gain in diagnostic quality that clinicians would prioritize over bandwidth savings. We will reframe the efficiency discussion accordingly in the revision.
>
> **W3: Sensitivity of synthetic data amount on FL results**
>
> We agree that the effect of synthetic data amount should be analyzed, and we address it from two perspectives.
>
> (1) Slide count — $O2O$ design. The ablation study (Table 2) compares one-to-one distillation (one synthetic slide per real slide, $N_{syn} = N_{real}$) against a compressed representation (10 synthetic slides per dataset). $O2O$ consistently outperforms the compressed setting across all datasets, suggesting that excessive compression loses inter-slide variation and weakens the slide-level correspondence used by the MIL pipeline.
>
> (2) Patch count per slide — $T$ ablation. Appendix A.2 (Fig. 5, upper) varies $T$ from $50$ to $5000$: AUC improves consistently from $T=50$ to $T=1000$, then plateaus and slightly degrades at $T=5000$. This supports $T=1000$ as a robust trade-off across all three datasets.
>
> Together, these two ablations confirm that both the slide count ($O2O$) and patch count ($T=1000$) are well-justified. We will make both analyses more prominent in the revision.
>
> We hope the clarifications above address the reviewer's concerns, and we will incorporate these points in the revision.
>
> **References**
> 1. Tian et al., Communication-Efficient Federated Skin Lesion Classification with Generalizable Dataset Distillation, MICCAI, 2023.
> 2. Song et al., Morphological Prototyping for Unsupervised Slide Representation Learning in Computational Pathology, CVPR, 2024.

---

> > ### Author Rebuttal · Reviewer_pvP8 · 2026-04-02
> >
> > Thanks for the response. My concerns are well addressed and I raise my score.

---

> > > ### Author Response · Authors · 2026-04-03
> > >
> > > Thanks for recognition and for raising score. We appreciate your time and effort in reviewing our paper.

---

### Decision · Program_Chairs · 2026-04-30

**Decision:**

Accept (regular)

**Comment:**

This paper addresses an important and practical problem in federated computational pathology: collaboration under heterogeneous local WSI models where standard weight aggregation is not feasible. Their rebuttal was exemplary—running new baselines, conducting adaptive mixture weight ablations, correcting errors transparently, and adding an entirely new survival prediction dataset (TCGA-KIRC). The dissenting reviewer's remaining concerns are either peripheral to the core contribution or have been comprehensively addressed by the new experiments. The paper is technically solid, has good clinical utility, and clears the bar for acceptance at ICML.